# The behavior of high-CAPE summer convection in large-domain large-eddy simulations with ICON

**Harald Rybka**[1], **Ulrike Burkhardt**[2], **Martin Köhler**[1], **Ioanna Arka**[2], **Luca Bugliaro**[2], **Ulrich Görsdorf**[3], **Ákos Horváth**[4], **Catrin I. Meyer**[5], **Jens Reichardt**[3], **Axel Seifert**[1], **and Johan Strandgren**[2]

[1]German Meteorological Service, Offenbach am Main, Germany
[2]Deutsches Zentrum für Luft- und Raumfahrt, Institut für Physik der Atmosphäre, Oberpfaffenhofen, Germany
[3]German Meteorological Service, Lindenberg, Germany
[4]Universität Hamburg, Germany
[5]Jülich Supercomputing Centre, Forschungszentrum Jülich, Jülich, Germany

**Correspondence:** Harald Rybka (harald.rybka@dwd.de)

**Abstract.** Current state of the art regional numerical weather prediction (NWP) models employ kilometre scale horizontal grid resolutions thereby simulating convection within its grey-zone. Increasing resolution leads to resolving the 3D
motion field and has been shown to improve the representation of clouds and precipitation. Using a hectometer-scale model in forecasting mode on a large domain therefore offers a chance to study processes that require the simulation of the 3D motion field at small horizontal scales, such as deep sum-
mertime moist convection, a notorious problem in NWP.

We use the Icosahedral Nonhydrostatic weather and climate model in large-eddy simulation mode (ICON-LEM) to simulate deep moist convection distinguishing between scattered, large scale dynamically forced and frontal convection. We
use different ground and satellite based observational data sets, that supply information on ice water content and path, ice cloud cover and cloud top height on a similar scale as the simulations, in order to evaluate and constrain our model simulations.

We find that the timing and geometric extent of the convectively generated cloud shield agrees well with observations while the life time of the convective anvil was, at least in one case, significantly overestimated. Given the large uncertainties of individual ice water path observations, we use a
suite of observations in order to better constrain the simulations. ICON-LEM simulates cloud ice water path that lies in-between the different observational data sets but simulations appear to be biased towards a large frozen water path (all frozen hydrometeors). The bias in frozen water path and
the longevity of the anvil are little affected by modifications

of parameters within the microphysical scheme. In particular one of our convective days appeared to be very sensitive to the initial and boundary conditions which had a large impact on the convective triggering, but little impact on the high frozen water path and long anvil life time bias. Based on this
limited set of sensitivity experiments, the evolution of locally forced convection appears to depend more on the uncertainty of the large-scale dynamical state based on data assimilation than of microphysical parameters.

Overall, we judge ICON-LEM simulations of deep moist
convection to be very close to observations regarding timing, geometrical structure and cloud ice water path of the convective anvil, but other frozen hydrometeors, in particular graupel, are likely overestimated. Therefore, ICON-LEM supplies important information for weather forecasting and
forms a good basis for parameterization development based on physical processes or machine learning.

## 1 Introduction

Regional km-scale weather forecasting is now routine in many numerical weather prediction (NWP) centers. Exam-
50 ples are the meteorological services of Switzerland, France, USA, United Kingdom, South Korea, Japan, Germany and China, who employ models with resolutions of 1.1 to 3 km in ascending order (see WGNE table at wgne.meteoinfo.ru for 2020). These regional NWP systems provide valuable guid-
55 ance for heavy precipitation and wind storm warnings, aircraft support, wind and solar power utilities as well as short

term prediction of typical near-surface and upper air variables.

Models at a resolution of 1–3 km describe convection within its grey-zone. They generally lack a direct treatment of deep convection, but still use shallow convection parametrizations. Permitting, but not fully resolving, deep convection forces the model developer to optimise either surface parameters of temperature and moisture or precipitation, one being the trigger of the other. Tuning (e.g. reduced mixing length) might for example be selected in a way to increase triggering of convection to yield a better precipitation peak earlier in the diurnal cycle by accepting biases in 2 m temperature (Baldauf et al., 2011; Hanley et al., 2015). More advanced approaches such as Arakawa and Wu (2013) and the blending approach of the Met Office (Boutle et al., 2014) are starting to be explored. The former employs a non-zero variable cumulus updraft fraction $\sigma$ and the latter calculates the turbulent length scale from the weighted average of a 1D turbulence model and a 3D Smagorinsky formulation. Those tuning challenges highlight the big gains that result from increasing resolution even further in order to resolve convection.

Lower resolution models (10–100 km or more), such as those used for global NWP or climate, on the other hand, struggle to simulate convection and its impact on the upper tropospheric water budget accurately; processes that are crucial for simulating important climate feedbacks (Bony et al., 2016) or regional precipitation responses (Stevens and Bony, 2013). In order to decrease the uncertainty in equilibrium climate sensitivity and feedbacks, the representation of such processes needs to be improved. Furthermore, progress in simulating the tropospheric water budget is key for estimating the impact of anthropogenic changes to cloudiness and climate.

Cloud resolving, as opposed to convection permitting, modeling is seen at present as a way of developing and testing parameterizations for low resolution models (Guichard and Couvreux, 2017; Gentine et al., 2018; Derbyshire et al., 2004), which require a detailed evaluation of the simulated cloud cover, water content, and cloud top heights. Cloud resolving modeling has been shown to lead to significant improvements in the representation of cloud and precipitation processes (e.g. Stevens et al., 2020; Khairoutdinov et al., 2009) and the continuing development of the models will improve the inclusion of small-scale couplings such as between turbulence and microphysics and with the land-surface (Guichard and Couvreux, 2017). Moreover, these models are starting to be run globally and have the potential to overcome the persistent problems of low-resolution models (Tomita et al., 2005; Satoh et al., 2019; Stevens et al., 2019).

Various model experiments have already been performed focusing on the realistic simulation of mid-latitude summer and tropical convection, encompassing different domain sizes and resolutions with the aim to aid parameterization development within low resolution models or to improve weather forecasts. Two are listed below.

– CASCADE: UK high-resolution modeling project to study organized convection in the tropical atmosphere using large domain cloud system resolving simulations (Holloway et al., 2013). The Unified Model (UM) at horizontal resolutions of 1.5 to 40 km was used for Africa, the Indian Ocean, and the West Pacific Ocean.

– The Convective Precipitation Experiment (COPE) field campaign (Leon et al., 2016) investigated the origins of heavy precipitation in the Southwestern United Kingdom during the summer of 2013. Simulations were run at resolutions of 1500 m, 500 m, 200 m, and 100 m using a nested setup of the UM.

The High Definition Clouds and Precipitation for Advancing Climate Prediction (HD(CP)$^2$) project demonstrated forecasting of clouds and precipitation on a 100 m scale over a large domain and realistic surface and boundary conditions. The framework used the ICOsahedral Non-hydrostatic (ICON) model (Zängl et al., 2015) further developed as a large-eddy model (Dipankar et al., 2015; Heinze et al., 2017) to perform these simulations, hereafter referred to as ICON-LEM (ICON Large-Eddy Model). Stevens et al. (2020) gave a general overview of HD(CP)$^2$ model simulations evaluated against a multitudeof observations, highlighting where horizontal resolution of O(100–1000 m) yields "added value" compared to climate model resolution. Improvements were found in particular regarding the location, propagation, and diurnal cycle of precipitation and clouds as well as the vertical structure of cloud properties. More specific topics within this project that have been covered, using ICON as a large-eddy model, are: arctic mixed-phase clouds (Schemann and Ebell, 2020), radiative effects of low-level clouds (Barlakas et al., 2020), diurnal cycle of trade wind cumuli (Vial et al., 2019), representation of Mediterranean tropical-like cyclones (Cioni et al., 2018), vertical-mixing of nocturnal low-level clouds (van Stratum and Stevens, 2018), aerosol-cloud interactions (Costa-Surós et al., 2020), convective organization or self aggregation (Pscheidt et al., 2019; Beydoun and Hoose, 2019; Moseley et al., 2020), soil moisture effects on diurnal convection (Cioni and Hohenegger, 2017), and using ICON at a lower storm resolving resolution, studying the spatial statistics of deep tropical convection (Senf et al., 2018). In this paper we use the unique capabilities of the HD(CP)$^2$ system to simulate realistic summer convective situations over land, where large amounts of convective available potential energy (CAPE) builds up during the course of the diurnal cycle, as a tool to study the evolution of a convective system and the skill of the model simulating that system and to investigate the uncertainty of forecasting such events.

The difficulty to predict precipitation location and amount arises to a large degree from the non-linearity originat-

ing from convective instability. Underlining that, Keil et al. (2014) established that predictability of convective precipitation depends on the convective adjustment time-scale, with higher predictability during strong large-scale forcing. Further, using a convection permitting model covering a large domain, Selz and Craig (2015) demonstrated that initial error growth is largest where precipitation rate is large. Initial error growth in the first hour transitions to large-scale perturbations on a 12 h time-scale. Moreover, resolutions of O(100 m) are necessary to realistically resolve and reproduce deep moist convection (Bryan et al., 2003).

Given the difficulties in predicting the triggering of convection under wide-spread CAPE and moderate westerly advection, sensitivities to the large-scale forcing and microphysics, as a key player in the physics of moist convection, are explored. We aim at evaluating ICON-LEM simulations regarding the water input into the upper troposphere due to summertime moist convection and the temporal evolution of the resulting anvil cloud. We employ a number of remote sensing products exploring whether our simulations of moist deep convection and their impact on the ice cloud field can be constrained by observations. Given the verification against a collection of observational data sets, we aim to arrive with a tool to investigate the uncertainty of convection. The different wavelengths used for observational estimates results in a spread that can be compared with forecast uncertainty from ICON-LEM sensitivity experiments.

To that effect, we use boundary and initial conditions from three operational NWP systems: COnsortium for Small-scale MOdeling (COSMO) at 2.8 km, ICON at 13 km, and Integrated Forecast System (IFS) at 16 km. Because the boundary and initial conditions are from short forecasts close to the analysis time, one might expect little impact on the ICON-LEM simulations. Additionally, we use the sensitivities to the choices within the cloud microphysics parametrization, such as ice particle shape, to explore the sensitivity to model error. In the literature one can find numerous studies of the sensitivity of convective storms and tropical cyclones to cloud microphysics (Wang, 2002; Milbrandt and Yau, 2006; Li et al., 2009; Van Weverberg et al., 2012; Bryan and Morrison, 2012, among many others). Most of them report significant sensitivity especially through the impact of evaporation and melting on the strength of the cold pool. Those sensitivity experiments are important for understanding the uncertainty connected with convectively generated precipitation and climate relevant aspects such as the longer term impact of convection on the upper tropospheric water budget.

To investigate the uncertainty of convection in high-CAPE weather situations, we first select several summer convective events over Germany that feature (i) strong and deep convective cells with little advection (e.g. 4 July 2015 extending into 5 July 2015), (ii) large convective cells connected with frontal passages (e.g. 20 June 2013 and 5 July 2015), and (iii) small scale scattered convective systems (e.g. 3 June 2016),

which are then simulated at 150 m resolution. See Table 1 for a list of all considered days.

To evaluate the performance of the control and sensitivity simulations of summer continental convection, we use ground-based and satellite observations from polar orbiting and geostationary sensors. To assess the quality of the high resolution simulations we rely on a suite of satellite ice water path (IWP) products representing the range of uncertainty in state-of-the-art retrievals. Furthermore, cloud ice water content (IWC), cloud top height (CTH) and an instrument-like ice cloud cover (ICC) conclude the evaluation of deep convective clouds.

The challenge to provide a meaningful comparison of cloud ice related quantities with spaceborne observations was reported in Waliser et al. (2009). Several follow-up studies (Eliasson et al., 2011; Waliser et al., 2011; Stein et al., 2011; Li et al., 2012; Eliasson et al., 2013; Li et al., 2016; Duncan and Eriksson, 2018) discussed the importance of considering the uncertainties in satellite IWP observations and their limitations for model evaluation. In order to analyze simulated cloud ice, it is necessary to know the unavoidable constraints of satellite observations. These range from retrieval sensitivities to microphysical assumptions (Yang et al., 2013), spatial and temporal sampling characteristics (Eliasson et al., 2013) and ultimately limitations that are determined by instrument type (active or passive sensors). This study uses a suite of observational data sets that reflects a realistic range of retrieval uncertainties for constraining the simulated cloud ice. These data sets encompass passive optical observations with high temporal resolution by the Meteosat Second Generation (MSG) satellite as well as with high spatial resolution by polar orbiting platforms. To explicitly show uncertainties of satellite ice products, different retrieval results are shown. In addition, a passive microwave sensor is also considered to complement the optical instruments.

The structure of the paper is as follows. Section 2 gives a synoptic overview of the selected cases to describe the meteorological background of the convective events. We describe the model simulations and the observations used for verification in Sect. 3 and 4. The evaluation of the ICON-LEM against observations is detailed in Sect. 5, while Sect. 6 describes the sensitivity studies for varying boundary and initial conditions and model physics before we conclude in Sect. 7.

## 2 Synoptic overview

Three summer days, 20 June 2013 and 4-5 July 2015, have been chosen to represent different high-CAPE summer convection types. In Fig. 1 snapshots of SEVIRI satellite images are juxtaposed with synthetic SEVIRI images for the respective days. The synthetic SEVIRI (Spinning Enhanced Visible and Infrared Imager) images were produced with RTTOV (Radiative Transfer for TOVS; Saunders et al., 1999, 2018), using as input ICON-LEM profiles of temperature, specific

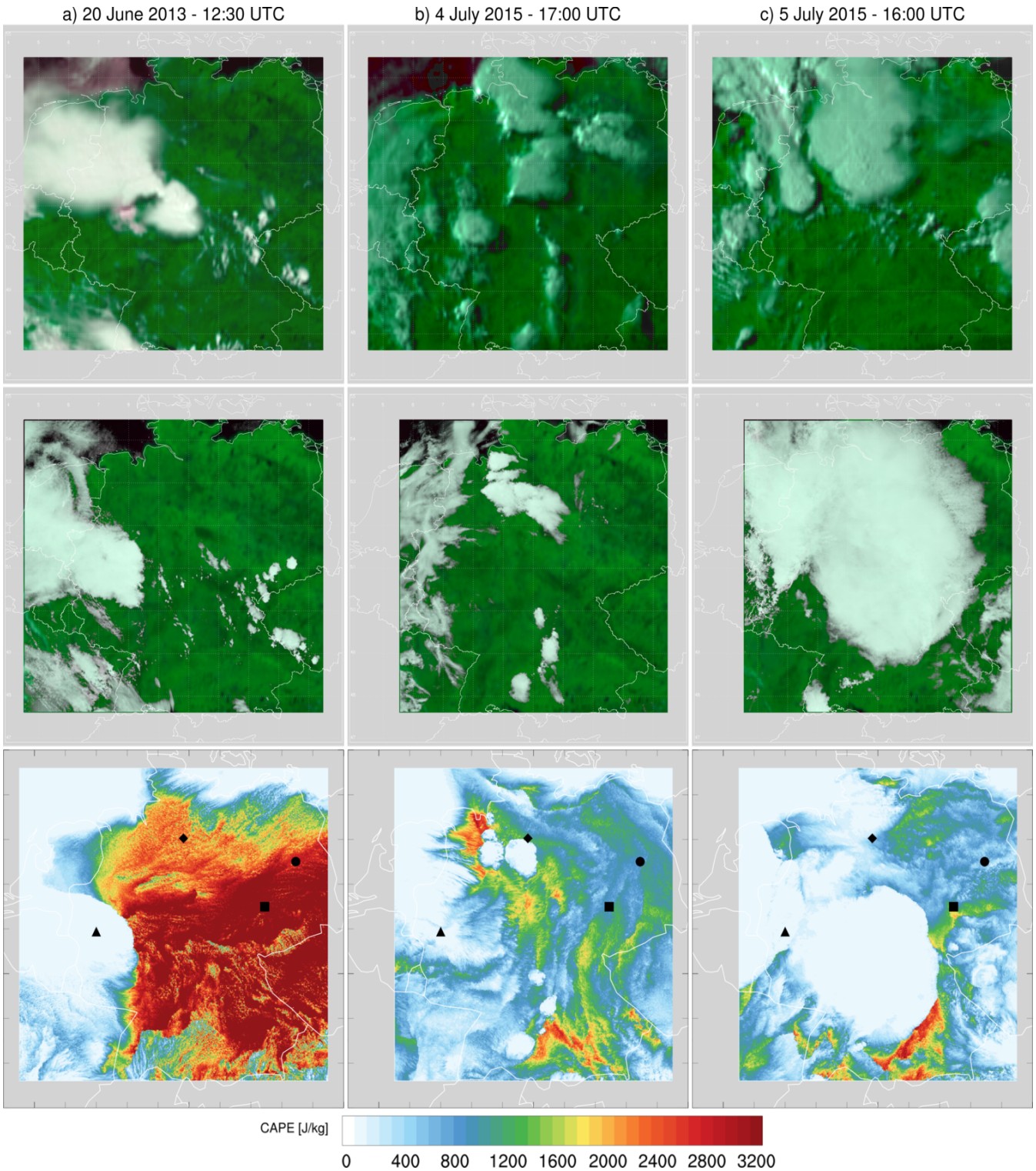

**Figure 1.** Synoptic situation as seen by SEVIRI for specific snapshots of the three selected days (upper row). Synthetic SEVIRI images of simulated cloud fields created with RTTOV are shown in the middle row. The false-color satellite images, both real and simulated, use the 0.6 micron reflectance for the red band, the 0.8 micron reflectance for the green band, and the average of the red and green bands for the blue band. Simulated CAPE values are displayed in the last row including the location of ground-based observational sites and initial release points of radiosondes: Bergen (diamond), Lindenberg (circle), Jülich (triangle) and Leipzig (square). SEVIRI images show the area from 47.6N to 54.5N and 4.5W to 14.5W. Due to a change in the model domain for the 4 and 5 July simulations the western border is shifted by one degree.

**Table 1.** Description of simulated convective days. Focus days analyzed in more detail in Sects. 2, 5.2 and 6 are marked in bold font

| simulation date | type of convection |
| --- | --- |
| **20 June 2013** | **highly organized frontal convection** |
| 29 July 2014 | scattered deep convection |
| **4 July 2015** | **large scale convective clusters** |
| **5 July 2015** | **convection embedded in front** |
| 29 May 2016 | strong convective phase with heavy rain and severe flooding in southern Germany |
| 3 June 2016 | scattered convection |
| 6 June 2016 | distinct diurnal cycle of convection |
| 22 June 2017 | strong convective phase with heavy rain |

humidity, cloud liquid water content (LWC) and cloud ice water content (IWC), as well as simulated surface skin temperature and 10 m wind speed. The ice optical properties come from the Baran parametrization (Vidot et al., 2015) and trace gas profiles were set to the RTTOV reference profiles. The RGB composites use the 0.6 micron reflectance for the red channel, the 0.8 micron reflectance for the green channel, and the average of the 0.6 micron and 0.8 micron reflectance for the blue channel. In addition, simulated CAPE values of ICON-LEM are displayed in the lowermost row in Fig. 1 for the respective time slices indicating atmospheric unstable regions.

The first selected day covers the evolution of a frontal zone on 20 June 2013. Germany lay between a ridge of an anticyclone spanning from the central Mediterranean Sea to the Baltics and a low pressure system in France. Organized convection developed all day along a convergence zone, predominantly in the western and northern part of Germany favored by hot surface temperatures above 35 °C under unstable atmospheric conditions. Radiosonde data from Lindenberg (Fig. 2a) point at high CAPE values and significant convective inhibition (CIN) over the east of the domain with a strong tropopause inversion at 190 hPa. Heavy rainfall including large hailstones above 5 cm has been reported for this day (https://eswd.eu/cgi-bin/eswd.cgi, last access: 20 October 2020; Dotzek et al. (2009)). Comparing the real and synthetic satellite images for 20 June 2013 in Fig. 1 (top and middle rows in column (a)) shows similar cloud structures around noon. The simulated CAPE field reflect huge potential of highly unstable regions (CAPE values over $3000 \, \mathrm{J \, kg^{-1}}$) above Germany. Based upon this single metric it can be seen that once convective inhibition is overcome, the potential to produce strong updrafts is given almost everywhere.

Furthermore, a 48 hour period starting at 0 UTC on 4 July 2015 has been chosen, which witnessed multiple local explosive convection cells on the first day and convection connected with a more synoptic scale frontogenesis on the second day (columns (b) and (c) in Fig. 1). For both days temperatures of nearly 40 °C have been registered, which support localized triggering of convection under unstable atmospheric conditions. Both criteria (high surface temperatures and unstable conditions in the lower and mid-troposphere) have been fulfilled on 4 July, leading to the formation of a couple of convective cells over the northern part of Germany. The radiosonde data from Bergen (Fig. 2b), very close to a convective cell, shows large CAPE values and close to no CIN with a strong tropopause inversion at 170 hPa. The development of these cells was quite explosive, resulting in a strong upward transport of moisture. Despite the convective region being highly localized, upper tropospheric detrainment of moisture and ice by deep convection created an extensive cirrus shield covering the complete northeastern part of Germany by the evening (not shown). Although the comparison of the observed and simulated cloud fields in Fig. 1b reveals structural differences, the overall ability of the model to simulate confined convective cells is clearly visible in the CAPE field. Circular white areas of consumed CAPE are located in the northern part of Germany surrounded by regions of higher CAPE.

The situation on 5 July is in the morning characterized by the decay of the large scale convective system of the previous day and later by a transition of a front aided by dynamical lifting induced by an upper air trough located over the North Sea. The satellite image in Fig. 1c shows the passage of the frontal system. The model produces an excessively large cloud structure that also extends too far south. Regions indicating very high CAPE are almost gone at 16 UTC with Bergen showing relative low values of CAPE (Fig 2c), but larger values above $1000 \, \mathrm{J \, kg^{-1}}$ occur over the northeastern part of Germany.

Each day presents a unique convective development, making these three cases an optimal test suite to study model performance under unstable atmospheric conditions.

## 3 Model and simulations

Simulations have been performed using the ICON modeling framework developed by the German Meteorological Service and the Max-Planck Institute for Meteorology (Zängl et al., 2015). Developments within HD(CP)$^2$ led to an ICON version specifically designed for regional to global large-eddy simulations (Dipankar et al., 2015). Several high-resolution model runs covering Germany with a grid mesh of 625 m have been carried out using realistic topography. Two additional one-way nested domains with 312 m and 156 m resolution are also embedded simultaneously in the model runs, using the lateral boundary conditions from the relative outer ones. The coarsest resolution (625 m) domain is referred to as DOM01, whereas the one with the finest grid size (156 m) is referred to as DOM03. Data of DOM02 (312 m horizontal resolution) is not used in this paper. The vertical model grid consists of 150 levels with layer thickness gradually increasing from 20 m in the lowermost model layer to 380 m

**Table 2.** Simulations with modified initial and lateral boundary conditions.

| simulation name | analysis | original resolution | frequency of analysis |
|---|---|---|---|
| ICON-LEM (default) | COSMO-DE | 2.8 km | 3 h |
| ICON-LEM lbc1 | ICON-NWP | 13 km | 12 h[*] |
| ICON-LEM lbc2 | IFS | 16 km | 12 h[*] |

[*] In-between analysis time steps forecasts were used as lateral boundary conditions.

at the top at 21 km in a height-based terrain-following co-ordinate system (Leuenberger et al., 2010). Using a model of hectometer scale over a huge domain inherently leads to resolved cloud dynamics; however, cloud microphysics, turbulence, and radiation still need to be parametrized.

A complete summary of the model setup and the physics package is given in (Heinze et al., 2017) and references therein. Here only the model aspects most relevant to this study are described. The following parametrizations have been used: A diagnostic Smagorinsky scheme with modifications by Lilly (1962) to account for subgrid-scale turbulence; An all-or-nothing approach for cloud cover neglecting subgrid-scale cloud fractions. The microphysical parametrization is based on Seifert and Beheng (2006a) applying a two-moment mixed-phase bulk scheme (SB scheme). Cloud condensation nuclei (CCN) concentration is prescribed as a function of pressure and vertical velocity (Hande et al., 2016). The CCN concentration decreases above 1500 m and is almost constant below. It represents typical aerosol conditions simulated with the COSMO-MUSCAT model (Multi-Scale Chemistry Aerosol Transport, Wolke et al., 2004, 2012). Ice nucleation is separated into a homogeneous and heterogeneous part. Homogeneous freezing follows the description of Kärcher and Lohmann (2002) and Kärcher et al. (2006), whereas the amount of heterogeneously nucleated ice particles is based on mineral dust concentrations as described in Hande et al. (2015). The Rapid Radiative Transfer Model (Mlawer et al., 1997) is used for radiative transfer calculations.

Model runs of 24 hours starting at 0 UTC have been performed to investigate the ability of a high-resolution cutting-edge model to forecast convective systems, especially to reproduce atmospheric ice composition.

The default ICON-LEM setup uses an initialization interpolated from the 2.8 km COSMO-DE (Baldauf et al., 2011) analysis of the German operational numerical weather model. Moreover, 3-hourly COSMO-DE analysis is used to relax ICON-LEM at the lateral boundaries using a 20 km nudging zone and COSMO-DE forecasts every hour in between. Unless stated otherwise, the DOM03 simulations used this setup.

In addition to the three days of interest described in Sect. 2, we further analyze five additional high-CAPE summer convection days, including small scale scattered convection (Table 1). These cases are analyzed in a statistical manner to-

gether with the three focus days in section 5.3, which summarizes the overall performance of ICON-LEM to represent atmospheric ice quantities in connection with deep convection.

Several sensitivity experiments have been conducted. The first set of additional simulations investigate the dependence of model performance on the initial and lateral boundary conditions (lbc). Two additional analyses from ICON-NWP (using the forecast system of DWD based on ICON) and IFS (cycle 41r1) models with lower spatial resolution (Table 2) have been remapped onto the ICON-LEM grid in order to initialize and force the high-resolution model during runtime. The temporal update of the lateral boundary forcing is the same for all three cases. The only difference for IFS and ICON-NWP forcing is that in between analysis time steps 3-hourly forecasts are available as boundary conditions (Table 2). Using different/coarser analysis allows us to address the sensitivity of ICON-LEM to large-scale forcing. Because ICON was made operational at DWD in 2015, this analysis has only been performed for the 4-5 July 2015 case (Sect. 6.1).

A second set of sensitivity experiments deals with changes to the two-moment microphysics scheme of Seifert and Beheng (2006a, b) (Appendix A4; Tab. A1). The prognostic variables within the SB scheme consist of the particle number concentration and mass mixing ratio of six different hydrometeor categories, namely cloud water, rain and four ice crystal classes: cloud ice, snow, graupel, and hail. The specific type or geometry of a frozen hydrometeor is referred in the following to as habit. We focus on the sensitivity simulations connected with ice crystal properties. In order to account for different ice crystal geometries and associated fall velocities based on Heymsfield and Kajikawa (1987), two separate simulations have been performed specifying cloud ice as hexagonal plates (simulation: 'hexPlate') or dendrites (simulation: 'dendrite'), both of which have lower terminal fall velocities compared to the default setup. A further sensitivity experiment, named 'stickLFOhigh', explores the impact of increased sticking efficiencies during ice hydrometeor collisions (snow-snow, ice-ice, snow-ice and graupel-snow) using parameters from Lin et al. (1983). The modified coefficients for the different sensitivity experiments are shown in Table 3. These simulations have been performed on the coarsest model grid of 625 m (DOM01). All microphysical sensitivity studies correspond to the 5 July 2015 case and

**Table 3.** Power law coefficients for the maximum diameter $D$ and terminal fall velocity $v$ of particles with mass $m$ as well as parameters determining the temperature ($T$) dependent sticking efficiency $E_{\text{stick}}(T)$ of ice hydrometeor collisions used in the microphysical sensitivity simulations.

| simulation name | $a$ (m kg$^{-b}$) | $b$ | $\alpha$ (m s$^{-1}$ kg$^{-\beta}$) | $\beta$ | $\gamma$ | $c_{\text{eff}}$ |
|---|---|---|---|---|---|---|
| ICON-LEM (DOM01) | 0.835 | 0.390 | 27.7 | 0.216 | 0.4 | 0.09 |
| hexPlate | 0.220 | 0.302 | 41.9 | 0.260 | 0.4 | 0.09 |
| dendrite | 5.170 | 0.437 | 11.0 | 0.210 | 0.4 | 0.09 |
| stickLFOhigh | 0.835 | 0.390 | 27.7 | 0.216 | 0.4 | 0.025 |

$D(m) \cong am^b$

$v(m) \cong \alpha m^\beta \left(\frac{\rho_0}{\rho}\right)^\gamma$, with density $\rho$ and surface density $\rho_0 = 1.225$ kg m$^{-3}$

$E_{\text{stick}}(T) = \exp(c_{\text{eff}}(T - T_3))$, with freezing temperature $T_3 = 273.15\ K$

are discussed in Sect. 6.2. Only for these microphysical sensitivity studies we make use of an explicit coupling of the two-moment microphysics scheme with radiation by calculating the effective radii of cloud ice and cloud droplet based on the predicted mass and number densities and the assumed particle size distribution.

## 4 Observational methods and data sets

We use ground-based as well as satellite-based observations to evaluate our simulations. Several previous studies have stated the differing magnitude and sampling characteristics of satellite-observed IWP or IWC (Waliser et al., 2009; Eliasson et al., 2011; Hong and Liu, 2015; Duncan and Eriksson, 2018). In evaluating the vertical and temporal distribution of simulated atmospheric ice in terms of IWP or IWC it is crucial to use multiple observational data sets representing a range of algorithms in order to estimate retrieval errors and uncertainties. For that reason, model simulations are compared to eight different observational methods, each of which has its own advantages and limitations.

For a vertically resolved point-to-point evaluation of the simulations at different sites, two ground-based observations have been taken into account:

- RAMSES (Raman lidar for atmospheric moisture sensing, Reichardt et al., 2012)

- Cloudnet retrievals (Illingworth et al., 2007)

For full-domain model evaluation, ice cloud properties from six different satellite retrieval algorithms are considered:

- SEVIRI CiPS (Cirrus Properties from SEVIRI, Strandgren et al., 2017a)

- SEVIRI SatCORPS (The Satellite ClOud and Radiation Property retrieval System, Minnis et al., 2008, Trepte et al., 2019)

- SEVIRI APICS (Algorithm for the Physical Investigation of Clouds with SEVIRI, Bugliaro et al., 2011)

- SEVIRI CPP (Cloud Physical Properties from SEVIRI, Roebeling et al., 2006)

- MODIS C6 (Moderate Resolution Imaging Spectroradiometer Collection 6 Cloud Products, Platnick et al., 2017)

- SPARE-ICE (Synergistic Passive Atmospheric Retrieval Experiment-ICE, Holl et al., 2014)

Four of them provide ice cloud properties with 15 min temporal resolution from the 12-channel SEVIRI imager aboard the geostationary MSG satellites (Schmetz et al., 2002), while two of them are from polar orbiting satellites (see next subsections for details). The different methods and characteristics of the observational data sets are described in the following.

## 4.1 RAMSES

RAMSES is the operational high-performance multiparameter Raman lidar at the Lindenberg Meteorological Observatory (Reichardt et al., 2012). It is equipped with a water Raman spectrometer (Reichardt, 2014) that facilitates direct measurements of cloud water content (CWC) on a routine basis. It is thus well suited for cloud microphysical studies, or for evaluating cloud models or the cloud data products of other instruments. However, such CWC measurements are only possible at night, under favorable atmospheric conditions and often only in the lower cloud ranges, because the Raman return signals from clouds are extremely weak, which makes them particularly vulnerable to background light and light extinction. For cirrus clouds it was possible to overcome this limitation by developing a retrieval technique which allows to estimate IWC under all measurement conditions (see Appendix A1 and Fig. A1 for more details). The new method was applied in conjunction with this case study of 4-5 July 2015 in Sect. 5.2.

## 4.2 Cloudnet

The ground-based data set of Cloudnet provides synergistic products from 35 GHz cloud radar, ceilometer, and multi-frequency microwave radiometer measurements. These products are derived for the observation sites Jülich, Leipzig, and Lindenberg using the same retrieval package developed in Cloudnet (Illingworth et al., 2007). Measurements are performed day and night, data are provided with a temporal and vertical resolution of 30 s and 60 m, respectively. Due to the low attenuation of the radar signals at this wavelength in the cloudy atmosphere, the clouds are detected almost in their entire vertical extent depending on the radar sensitivity. Only in situations with strong precipitation the attenuation is higher and thus the cloud detection capability lower.

As the first step, the retrieval performs a target classification including the determination of cloud base and top. Radar profiles of reflectivity, Doppler velocity, and ceilometer backscatter profiles are used for this purpose, as well as temperature and humidity profiles provided by a NWP model (e.g. COSMO-DE for Lindenberg) or radiosoundings. Vertical profiles of LWC and IWC are derived subsequently. For echoes classified as ice, IWC is calculated from radar reflectivity and temperature using an empirical formula, which was derived on the basis of a large mid-latitude aircraft data set (Hogan et al., 2006). The random error of the IWC retrieval is approximately between +50 % and -33 % for IWC values in the range of 0.03 and 1 g m$^{-3}$. A potential systematic error in IWC, which is mainly caused by systematic errors in radar reflectivity, is of the same order of magnitude assuming a radar calibration error of 2 dBZ. It should also be noted that due to the limited sensitivity of the cloud radar, very thin clouds (with small ice crystals) may not be detected.

## 4.3 SEVIRI CiPS

The Cirrus Properties from SEVIRI (CiPS Strandgren et al., 2017a) algorithm detects cirrus clouds and retrieves their cloud top height (CTH), ice optical thickness ($\tau$), and IWP using thermal observations from MSG/SEVIRI. To this end, a set of neural networks trained with SEVIRI observations and coincident cirrus properties retrieved with the Cloud-Aerosol LIdar with Orthogonal Polarization (CALIOP) instrument (Winker et al., 2009) are used. Day and night coverage, a temporal resolution of up to 5 min, and a spatial resolution of 3 km at nadir, makes the algorithm ideal for evaluating the temporal evolution of high cloud fields. CiPS targets thin cirrus clouds, detecting, compared to CALIOP, about 50, 60, and 80 % of cirrus clouds with an ice optical thickness of at least 0.05, 0.08, and 0.14 (Strandgren et al., 2017a), which corresponds to an IWP of roughly 0.6, 1.0, and 3.0 g m$^{-2}$, respectively. The CTH retrieved by CiPS has an average error of 10 % or less for cirrus clouds with a top height greater than 8 km, again with respect to CALIOP observations over the entire MSG disk. When looking at the geographic distribution of CTH accuracy of CiPS versus CALIOP, it turns out that the CiPS neural network has a mean percentage error very close to zero in Germany for ice clouds located between 8 and 11 km. For lower clouds, CiPS tends to overestimate and for higher clouds to underestimate CTH. The high sensitivity of CiPS to thin cirrus does, however, lead to a quick saturation of the IWP and $\tau$ retrievals in thicker cirrus clouds. Maximum IWP and $\tau$ amount to approximately 100 g m$^{-2}$ and 4, respectively. This makes the algorithm unsuitable for the evaluation of modeled IWP in this paper, where thick convective clouds are analysed, but CiPS is an ideal tool to study e.g. the spatial extent of anvil cirrus from the convective outflow including the optically thinner cloud edges.

## 4.4 SEVIRI APICS

The Algorithm for the Physical Investigation of Clouds with SEVIRI (APICS, Bugliaro et al., 2011) computes optical thickness $\tau$ and ice crystal effective radius $r_{\mathrm{eff}}$ for pixels identified as cirrus by CiPS, by means of the Nakajima-King method (Nakajima and King, 1990) using two SEVIRI solar channels centred at 0.6 and 1.6 $\mu$m. IWP is derived from these two quantities ($\tau$, $r_{\mathrm{eff}}$) under the assumption of a vertically homogeneous cloud layer using the relationship IWP $= 2/3\rho_{\mathrm{ice}}r_{\mathrm{eff}}\tau$, where $\rho_{\mathrm{ice}} = 917$ kg m$^{-3}$ is the density of ice. The algorithm assumes the general ice crystal shape mixture from Baum et al. (2011). Retrieved optical thickness is up to 200, while effective radius is between 5 and 60 $\mu$m, yielding a maximum retrieved IWP of $\approx 7300$ g m$^{-2}$. In contrast to CiPS, APICS is not limited to thin cirrus but is only available during daytime.

## 4.5 SEVIRI SatCORPS

The Satellite ClOud and Radiation Property retrieval System (SatCORPS) is a comprehensive set of algorithms designed to retrieve cloud micro- and macrophysical information day and night from meteorological satellite imager data. These algorithms were originally developed for the NASA Clouds and Radiant Energy Systems (CERES) project (Minnis et al., 2020, Trepte et al., 2019) and adapted for application to other polar-orbiting and geostationary imagers, including SEVIRI. Using radiances in the 0.6 $\mu$m (visible), 3.9 $\mu$m (shortwave-infrared), 10.8 $\mu$m (infrared), and 12.0 $\mu$m (split-window) bands, three different methods are employed depending on time of day and cloud opacity to retrieve cloud optical thickness ($\tau$), ice crystal effective diameter ($D_{\mathrm{eff}} = 2r_{\mathrm{eff}}$), and cloud effective temperature ($T_c$).

During daytime, the Visible Infrared Shortwave-infrared Split-window Technique (VISST) uses the visible, shortwave-infrared, and infrared radiances to determine $\tau$, $D_{\mathrm{eff}}$, and $T_c$, respectively, by an iterative process that also exploits the split-window band to aid in phase determination. The VISST is similar in essence to the classic Nakajima and King (1990) bispectral method.

For thin non-opaque cirrus ($\tau < 8$) during nighttime, the Shortwave-infrared Infrared Split-window Technique (SIST) retrieves the same parameters from brightness temperature differences between the shortwave-infrared and infrared bands and those between the infrared and split-window bands. The VISST/SIST reflectance lookup tables (LUTs) and emittance parametrizations are calculated for smooth solid hexagonal ice crystals. Assuming that the retrieved ice crystal effective diameter represents the average over the entire cloud thickness, IWP is computed from the following cubic equation:

$$
\begin{aligned}
\mathrm{IWP} = \tau\,(0.259\,\mathrm{D_{eff}} + 0.819 \times 10^{-3}\,D_{eff}^2 \\
- 0.880 \times 10^{-6}\,D_{eff}^3)
\end{aligned}
\tag{1}
$$

For thick opaque ice clouds ($\tau > 8$) during nighttime, the Ice Cloud Optical Depth from Infrared using a Neural network (ICODIN) method is used (Minnis et al., 2016), complementing the SIST applicable to semitransparent cirrus. The ICODIN retrieves $\tau$ and IWP by training shortwave-infrared, infrared, and split-window radiances against the CloudSat radar-only 2B-CWC-RO product (Austin et al., 2009), which includes vertical profiles of IWC and ice particle effective radius. The method can be used to derive ice cloud $\tau$ up to 150; however, $\tau$ and thus IWP for the deepest convective clouds is still frequently underestimated. According to equation (1), with a maximum $\tau$ of 150 and a maximum effective diameter of 150 $\mu$m, the maximum IWP that can be derived using this approach is $\approx 8100\,\mathrm{g\,m^{-2}}$. SatCORPS is the only geostationary retrieval used here that provides IWP during both day and night for thin and thick ice clouds. Note, however, that at the day-night transition, the weak solar component in the 3.9 $\mu$m band increases the uncertainty in the opaque vs. semitransparent cloud classification and can result in the use of default values for $\tau$ (16 or 32), which are significant underestimates in deep convective clouds (see the sudden dip in IWP around 18 UTC in Fig. 5). Nighttime retrievals are inherently more uncertain due to the reduced information content resulting from the lack of the solar reflectance channel (Minnis et al., 2020) and the nighttime algorithm has a tendency to favor ice-phase retrievals (Yost et al., 2020). The pixel-level 15-minute temporal resolution SEVIRI SatCORPS data were obtained from NASA Langley Research Center (http://satcorps.larc.nasa.gov, last access: 15 April 2019).

## 4.6 SEVIRI CPP

The Cloud Physical Properties (CPP) algorithm (Roebeling et al., 2006) is a bispectral method (Nakajima and King, 1990), which uses SEVIRI 0.6 $\mu$m and 1.6 $\mu$m solar reflectance measurements to retrieve cloud optical thickness and ice particle effective radius during daytime. The retrievals are based on LUTs of top-of-atmosphere reflectances calculated for plane-parallel layers of randomly oriented monodisperse roughened hexagonal ice crystals (Hess et al., 1998). Assuming no vertical variation in ice crystal size, the IWP is calculated as for APICS, although the density of ice is assumed to be $\rho_{\mathrm{ice}} = 930\,\mathrm{kg\,m^{-3}}$. Specifically, we use data from the CLoud property dAtAset using SEVIRI – edition 2 (CLAAS-2) archive provided by the EUMETSAT Satellite Application Facility on Climate Monitoring (Benas et al., 2017). The pixel-level IWP retrievals are available every 15 minutes at a spatial resolution of $\approx 6$ km over Germany. For this algorithm, maximum retrieved optical thickness and effective radius are 100 and 62.5 $\mu$m respectively, which result in a maximum IWP of $\approx 3900\,\mathrm{g\,m^{-2}}$. Due to the different assumed ice habit and smaller $\tau$ truncation threshold, SEVIRI CPP retrieves smaller IWP values than SEVIRI APICS, although the algorithms are otherwise very similar. Older versions of CPP and APICS also show in Bugliaro et al. (2011) that they provide similar results, with again CPP producing lower values of optical thickness and IWP than APICS.

## 4.7 MODIS

MODIS is a 36-channel imager with spatial resolutions of 250, 500 or 1000 m at nadir and with a swath width of 2330 km. It is the key instrument aboard the Terra and Aqua NASA satellites and provides global coverage every 1 or 2 days. The MODIS cloud microphysical products are also obtained by the Nakajima and King (1990) bi-spectral method and provide daytime estimates of cloud optical thickness and ice particle effective radius from solar reflectances measured in a non-absorbing visible band and a water-absorbing near-infrared band (Platnick et al., 2017). Three different spectral cloud retrievals are performed by combining the 0.66 $\mu$m channel separately with the 1.6 $\mu$m, 2.1 $\mu$m, and 3.7 $\mu$m channel, although here we only use the primary 0.66 $\mu$m – 2.1 $\mu$m channel pair. In the latest Collection 6 algorithm, the plane-parallel reflectance LUTs are calculated for a single ice shape of severely roughened compact aggregates composed of eight solid columns. Assuming a vertically homogeneous cloud, the IWP is derived as for SEVIRI APICS and SEVIRI CPP. The 1 km resolution IWP retrievals are available twice a day from the Terra and Aqua satellites, which are in a 1030 Local Solar Time (LST) descending node and 1330 LST ascending node sun-synchronous polar orbit, respectively. Maximum retrieved optical thickness and effective radius are 100 and 60 $\mu$m, yielding a maximum retrieved IWP of $\approx 3700\,\mathrm{g\,m^{-2}}$. Benas et al. (2017) compared SEVIRI CPP and MODIS retrievals. They found lower CPP IWPs than MODIS IWPs, similar to our observations (see Fig. 5), mainly caused by lower CPP ice effective radius values.

## 4.8 SPARE-ICE

The Synergistic Passive Atmospheric Retrieval Experiment-ICE (SPARE-ICE) features a pair of artificial neural networks that use infrared and microwave radiances as input to

detect ice clouds and retrieve their IWP (Holl et al., 2014). The networks were trained by collocating AVHRR channel 3B, 4, 5 (3.7 $\mu$m, 10.8 $\mu$m, 12 $\mu$m) and MHS channel 3, 4, 5 (183$\pm$1 GHz, 183$\pm$3 GHz, 190 GHz) radiances with IWP retrievals from the CloudSat/CALIPSO radar-lidar synergy product 2C-ICE (Deng et al., 2010). The exclusion of solar reflectances from SPARE-ICE allows retrievals both day and night; however, the reliance on microwave measurements results in fairly large footprints varying from 16 km in diameter at nadir to $52 \times 27\,\text{km}^2$ in areas at the edge of the scan. The lower and upper sensitivity limits of SPARE-ICE are $10\,\text{g}\,\text{m}^{-2}$ and $O(10^4)\,\text{g}\,\text{m}^{-2}$, respectively, with the median fractional error between SPARE-ICE and 2C-ICE IWP being a factor of 2. For the current study, data are available from the MetOp-A/B (0930 LST descending node) and NOAA-18/19 (1500-1630 LST and 1330-1400 LST ascending node) satellite overpasses.

## 4.9 Interpretation of satellite IWP retrievals

Despite the wide variety of available satellite instruments (imagers, sounders, lidar, radar) and retrieval methods exploiting the information obtained with these instruments, determining atmospheric ice mass has been recognized as a great challenge for remote sensing (Waliser et al., 2009; Eliasson et al., 2011), which has seen only limited progress in the past decade as large discrepancies in IWP remain among satellite data sets (Duncan and Eriksson, 2018). In this context, "ice" represents all frozen hydrometeors, including the smaller suspended (or floating) cloud ice as well as the larger precipitating forms such as snow, graupel, and hail. Current satellite retrieval methods are unable to truly distinguish suspended ice from precipitating ice, which makes estimates from these techniques rather uncertain in thick, multi-layer, mixed-phase and mixed-habit cloud fields. The measured signal, and hence the derived ice mass, is a weighted sum of the individual contributions from the different ice habits. Habit weighting, however, varies by retrieval method and is poorly characterized if at all, which complicates model-satellite comparisons because the various satellite products all refer to "ice water path", without any qualifying caveats about their differing sensitivities. In turn, this also means that different instruments are sensitive to different ice cloud types (Eliasson et al., 2011) such that several space borne sensors are needed to cover the full range of ice clouds.

Passive VIS-NIR methods can derive IWP only indirectly, from optical thickness and effective particle size. However, they infer particle size from cloud-top measurements and usually provide an estimate of cloud-top ice particle size. Thus, they are unable to obtain information about ice particle sizes in lower layers inside vertically thick clouds and the used bulk IWP formulas that assume vertical homogeneity (see Sect. 4.4, 4.6 and 4.7) cannot a priori account for vertical variations in extended clouds.

Furthermore, these methods are subject to saturation effects (affecting normally a few percent of pixels in our analyzed scenes, mainly the convective cores; in situations with large scale convective activity many pixels may be affected e.g. 20% of pixels on the 20th June 2013), because visible reflectance loses sensitivity to optical thickness in thick clouds. As a result, the maximum reported optical thickness is truncated at a threshold value varying between 100–200 depending on the data product. The maximum reported ice particle effective radius also varies among data sets, although in a narrower range, depending on the ice optical properties used. In addition, the retrieved optical thickness and particle effective radius strongly depend on the assumed ice particle shape (smooth or roughened, solid or hollow, hexagonal columns or aggregates etc.), even for unsaturated input reflectances. For instance, Eichler et al. (2009) show that for thin ice clouds with an optical thickness between 3-5, the choice of ice particle shape leads to uncertainties of up to 70% for optical thickness and 20% for effective radius. Retrievals in deep convective clouds have uncertainties of similar magnitude or even larger. As a last source of uncertainty one has to mention that the passive optical retrievals assume the cloud to consist of either ice or liquid water clouds according to their cloud top phase. When in convective clouds both phases are present - liquid water in the lower and ice in the upper part, with a mixed phase layer in between - the retrieved IWP accounts in part for the liquid water layers and thus tends to overestimate the real IWP. However, the truncation of the retrieved optical thickness mentioned above partially compensates for this overestimation. Nevertheless, the combination of all the above effects can easily lead to a factor of 2–3 variation in the estimated domain-mean IWP. In our VIS-NIR satellite data, SEVIRI CPP shows the smallest IWPs and SEVIRI Sat-CORPS the largest ones, with SEVIRI APICS and MODIS values being in between (see Fig. 5), providing a broad range of estimates reflecting the current state-of-the-art.

The SPARE-ICE retrievals, on the other hand, were trained on CloudSat/CALIPSO active radar-lidar retrievals, whose sensitivity is markedly shifted to the larger ice hydrometeors. Therefore, SPARE-ICE usually provides the highest IWPs due to the inclusion of graupel and hail, although the Sat-CORPS passive VIS-NIR retrieval can occasionally produce IWPs of comparably large magnitude, as shown later.

As a last issue, the different spatial resolutions of the satellite measurements must be mentioned. Since MODIS provides the finest resolution, SEVIRI an intermediate resolution and SPARE-ICE the coarsest, MODIS is able to catch peaks of high IWP that are smoothed out in the other two observational data sets. However, the differences in instantaneous pixel-level estimates due to different spatial resolutions are largely reduced in domain-mean IWP.

In our model validation effort, we follow a somewhat qualitative rule of thumb recommended by Waliser et al. (2009) and consider the SEVIRI/MODIS passive VIS-NIR IWP retrievals as more representative of the smaller suspended

cloud ice mass and treat the SPARE-ICE radar/lidar-trained IWP retrievals as more indicative of the total ice mass (i.e. cloud plus precipitating ice).

## 4.10 Comparison to model simulations

When comparing vertical profiles of cloud hydrometeors from ICON-LEM to surface lidar (RAMSES, Sect. 4.1) or radar (Cloudnet, Sect. 4.2) observations, the model grid points nearest to the locations of ground based instruments are selected. Furthermore, we take into account the neighboring grid points since differences between observations and simulations may be easily explained in case of inhomogeneities. This comparison approach is intended to provide an assessment of the model simulation error considering potential temporal or spatial displacements.

When comparing model quantities with satellite observations, we proceed as follows. Ice cloud cover (ICC) and CTH are evaluated against CiPS retrievals (Sect. 4.3), which have a high detection efficiency for ice clouds, including thin ice clouds. In order to compare the CiPS results with modeled ICC and CTH, we need to consider the detection efficiency dependent on IWP or optical thickness of CiPS. We therefore calculate IWP from the simulated cloud fields and respectively apply cut-off values of 0.6 and $3.0\,\mathrm{g\,m^{-2}}$ corresponding to the 50% and 80% detection probability of CiPS (see Sect. 4.3). The resulting IWP is called $\mathrm{IWP_{CiPS-sim}}$ in the following. $\mathrm{IWP_{CiPS-sim}}$ of the simulated cloud field is calculated from IWC and LWC below -25 °C, because CiPS increasingly misidentifies supercooled liquid water as ice at lower temperatures (Strandgren et al., 2017b), and above -25 °C from IWC only if IWC is larger than LWC. If $\mathrm{IWP_{CiPS-sim}}$ does not exceed the threshold value cloud cover is set to 0.0. CTH in turn is set to the height where $\mathrm{IWP_{CiPS-sim}}$ first exceeds the threshold when integrating $\mathrm{IWP_{CiPS-sim}}$ from the top of the cloud layer. The ICC and CTH calculated for the two IWP thresholds give a measure for the uncertainty in the CiPS retrievals. Very thin simulated ice clouds ($\mathrm{IWP} < 0.6\,\mathrm{g\,m^{-2}}$) are neglected and the influence of mixed phase clouds are limited in our analyzed ICC and CTH. We note that the above CiPS-specific ICC should not be confused with the model's own output variables of high cloud cover or cirrus cloud cover, which are calculated differently.

IWP averaged over the whole simulation domain is compared to the satellite products from Sect. 4 to account for the uncertainty in IWP retrievals. The SatCORPS retrieval method switches input channels at sunset between 18 and 19 UTC (see Sect. 4.5), which leads to unreliable estimates around that time. Furthermore, two separate domain averaged IWP values are calculated from ICON-LEM data: one strictly for cloud ice water path (tqi) and one for total frozen water path (tqf). The former is the column integrated and domain averaged ice content (qi) of cloud ice crystals only, whereas tqf comprises all ice habits, including the larger agglomerates such as snow (qs), graupel (qg), and hail (qh) within the two-moment microphysics. Please refer to Sect. 4.9 for a discussion about the sensitivity of the single satellite retrievals to different ice classes.

## 5 Evaluation of ICON-LEM simulations against observations

We focus on ice cloud properties in the ICON-LEM simulations, which have until now been only evaluated in a lower resolution version of ICON in simulations over the equatorial Atlantic (Senf et al., 2019). More specifically, the impact of deep summertime convection on ice cloud properties is investigated over Germany. We focus on a few case studies (Sect. 2) and study the evolution of the convective outflow making use of radiosonde data, remote sensing data from ground based instruments and those on geostationary and polar orbiting satellites (Sect. 4).

## 5.1 Evaluation of simulated temperature profiles with radiosonde data

This section is dedicated to present a comparison of simulated thermodynamic profiles and radiosonde data for specific locations and times for each summertime convective event presented in Sect. 2. The comparison with model data provides a brief verification of the model setup and its ability to reproduce the stability and moisture profile and how conducive it is for deep convection including an indication of possible cloud top height. For this evaluation of temperature profiles observational radiosonde data, archived at the Climate Data Center of the German Weather Service (https://opendata.dwd.de/climate_environment/CDC/, last access 13 November 2020), has been used.

Figure 2 shows three different atmospheric profiles measured by radiosonde soundings presented in Skew-T/log-P diagrams. The location and time of ascent is stated above each panel and is closely matching the taken snapshots in Fig. 1. When comparing the model simulation with the radiosonde measurements the drift of the radiosonde during the ascent has been taken into consideration adjusting location, time and pressure altitude of the simulated profile in accordance with the drift. The red lines illustrate an undiluted air parcel ascent above the level of free convection and visualize the corresponding CAPE. CAPE values are given above each figure.

Figure 2a shows measured (black) and simulated (blue) profiles at Lindenberg for 20 June 2013 at 12 UTC. The comparison illustrates very similar temperature (solid lines) and dew-point temperature (dashed lines) profiles reflecting a high CAPE (red) environment. CIN is higher in the simulation than in observations which is dominated in both observations and simulations by an inversion layer of several K. A tropopause inversion is seen in the measured profiles, which is less sharply reproduced by ICON-LEM highlighting pos-

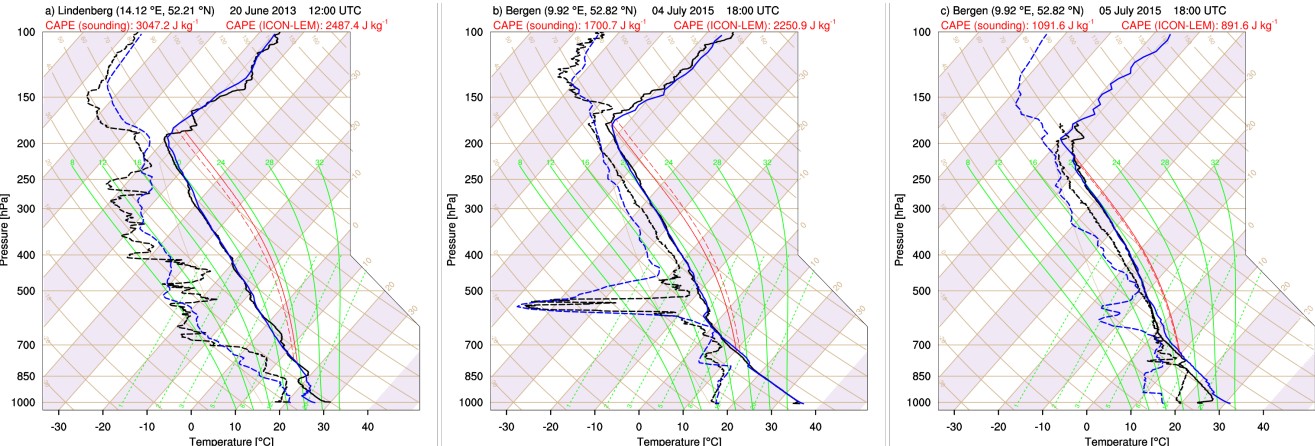

**Figure 2.** Comparison of vertical profiles plotted on a Skew-T/log-P diagram for the three simulated days. The location and start of ascent is given on top of each panel, approximately matching the point in time of the synoptic situations in Fig. 1. The sounding profile is depicted in black, whereas blue lines display simulated profiles (solid lines: temperature; dashed-lines: dew-point temperature). Unstable regions are highlighted with red lines illustrating CAPE values given on top of each panel (solid red lines: CAPE of the sounding; dashed red lines: simulated CAPE). All other basic lines are: isobars (in hPa; horizontal brown lines), isotherms (°C; solid brown lines sloping from the lower left to the upper right), dry adiabats (°C; slightly curved, solid brown lines sloping from the lower right to the upper left), saturation adiabats (°C; slightly curved, solid green lines) and saturation mixing ratios (g kg$^{-1}$; almost straight, dashed green lines starting from the lower left to the upper right).

sible higher cloud tops than observed. In the simulation the upper troposphere at around 200 hPa is ice saturated while observations indicate slightly lower relative humidity.

Explosive localized convective cells characterize the day of 4 July 2015. One of these cells was located in the near vicinity of Bergen, which happened to serve as a launching position of a radiosonde ascent. The corresponding profile is shown in Fig. 2b. The simulated dew-point (blue dashed) and temperature (blue solid line) profile closely follow the observed ascent up to 500 hPa reproducing the very dry layer at 550 hPa and very high surface temperatures (above 35°C). The mid-troposphere is slightly drier and the upper troposphere slightly moister in the model (between about 170 and 210 hPa ice saturation is reached in the simulations) while the tropopause level is identical in the simulation and observations. Focusing on the lower troposphere extremely low CIN (convective inhibition) values provide the potential for an explosive development of a convective cell. CAPE (red) is large in both the simulated (dashed) and observed (solid) profile.

The 5th July 2015 is dominated by the passage of a frontal system (compare Fig. 1c). Comparison of the radiosonde and simulated profiles Fig. 2c) show that in the simulations temperatures are lower below 750 hPa. This is consistent with an earlier passage of the front over Bergen in the simulations with a greater consumption of CAPE at this time. The whole atmosphere above 500 hPa is very moist reaching ice saturation between 240 and 190 hPa with the simulations slightly drier in the mid atmosphere. Whereas the level of the tropopause in ICON-LEM is around 190 hPa, the balloon

bursts at 170 hPa without providing a clear signal of the observed tropopause at this level.

We have limited the radiosonde comparison to the times of day depicted in Fig. 1 and the locations strongly affected by convection or showing large CAPE values. In total 40 profiles have been analysed of which 30 show similarly small discrepancies as in Fig. 2a and Fig. 2b with only few profiles exhibiting discrepancies that are as large as in Fig. 2c. Overall this comparison supports that ICON-LEM provides accurate results concerning thermodynamic states conducive to convection for the selected high-CAPE convective cases. The analysis of those 3 days indicates a possible bias consisting of a too weak tropopause inversion.

## 5.2 Evaluation of simulated ice cloud properties with remote sensing data

In this section, we evaluate the ability of ICON-LEM to simulate the convective outflow and its temporal evolution for the three large-scale summertime convective events over Germany that were introduced in Sect. 2.

### 5.2.1 Comparison to ground based measurements

First we use ground-based observations (Cloudnet and RAMSES, Sect. 4.2 and 4.1) to evaluate simulated ice water content for different locations in Germany. Figure 3 shows IWC meteograms for 20 June 2013, comparing three different Cloudnet sites with ICON-LEM. The comparison is performed at the model grid points nearest to the respective Cloudnet site, as already mentioned in Sect. 4.10.

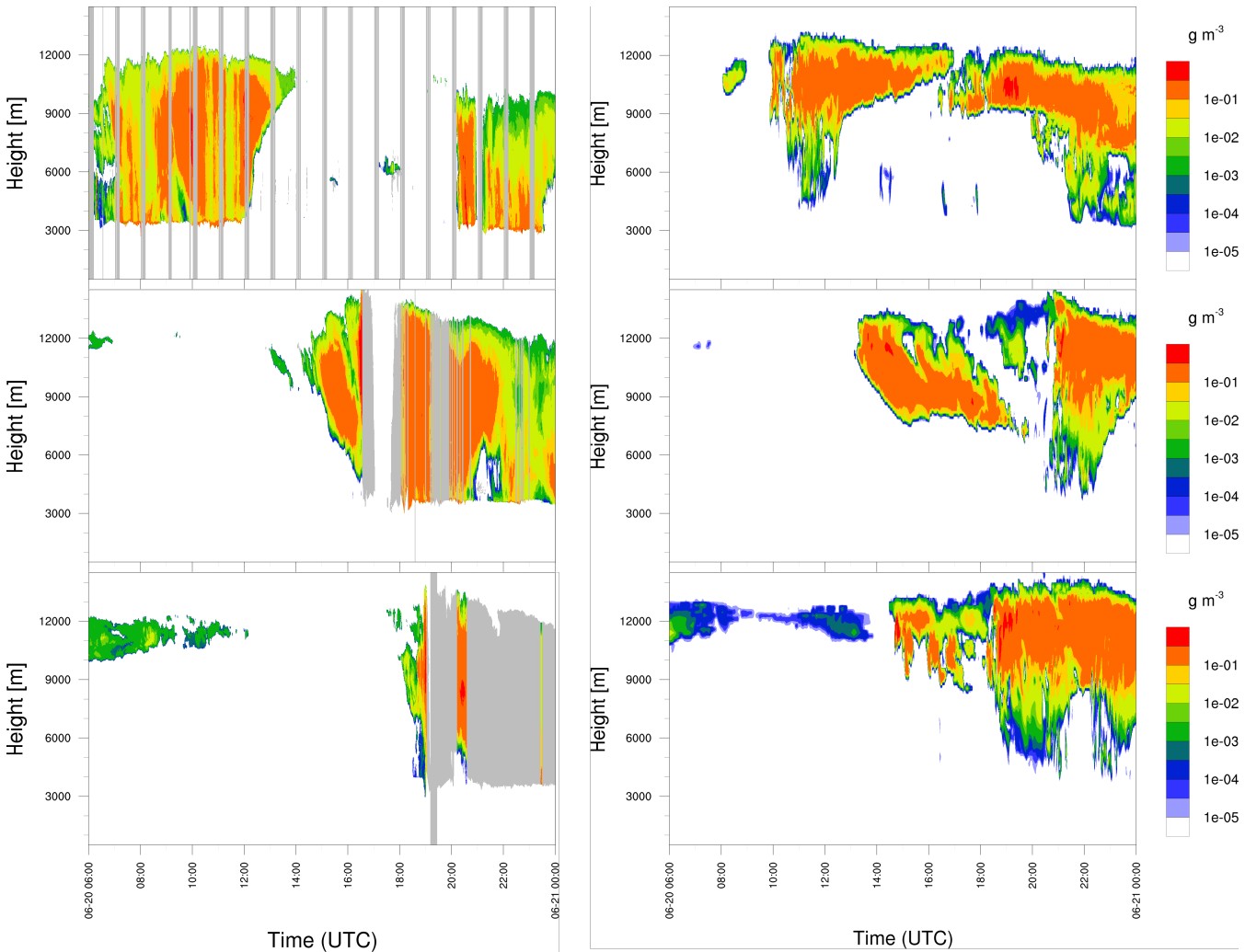

**Figure 3.** Temporal evolution of ice water content observed by Cloudnet with a $30\,\mathrm{s}$ temporal resolution (left) vs. simulated by ICON-LEM (right) for three stations: Jülich (top), Leipzig (middle) and Lindenberg (bottom) for 20 June 2013. Grey shaded areas indicate missing values within the Cloudnet data or points in time where the retrieval could not evaluate ice water content due to falling precipitation. The periodically reoccurring data gaps in the Jülich data are caused by a radar scan every hour in which the antenna is not vertically pointed and thus no Cloudnet retrieval is possible.

Comparing the overall magnitude of observed and simulated IWC shows that ICON-LEM is capable of providing a good estimate of high in-cloud IWC values ranging between $10^{-4}$ and $1\,\mathrm{g\,m^{-3}}$. Having a closer look at cloud edges, a transition to lower IWC values is visible in ICON-LEM, which corresponds well to the observed width of the decreasing ice water content at cloud edge. This indicates a good representation of cloud edge mixing by entrainment and detrainment processes. When comparing the cloud fields and in particular cloud top height, it should be taken into account that very low IWC values cannot be retrieved due to the limited sensitivity of the radars. The minimum retrievable IWC depends on radar parameters, height, and temperature. At 10 km altitude, for example, the smallest IWC which can be obtained is $1.8\ 10^{-4}\,\mathrm{g\,m^{-3}}$ for Lindenberg, $3.34\ 10^{-4}\,\mathrm{g\,m^{-3}}$

for Leipzig, and $3.44\ 10^{-3}\,\mathrm{g\,m^{-3}}$ for Jülich. The limited radar sensitivity likely contributes to the $500\,\mathrm{m}$ to $1000\,\mathrm{m}$ cloud top height bias in ICON-LEM simulations relative to observations. Therefore, an additional analysis has been performed in Sect. 5.2.2 (see Fig. 6) taking into account the detection efficiency of ice clouds as a function of ice water path.

It should be noted that no perfect agreement is expected in IWC development when comparing individual model grid points against ground-based observations. Nevertheless, the modeled cloud ice development, especially for Lindenberg and Leipzig, reveals a good description of the observed temporal evolution, including the representation of the cirrus layer over Lindenberg between 6:00 and 14:00 UTC.

For the second convective episode on 4-5 July 2015, no validation data are available from most of the Cloudnet sta-

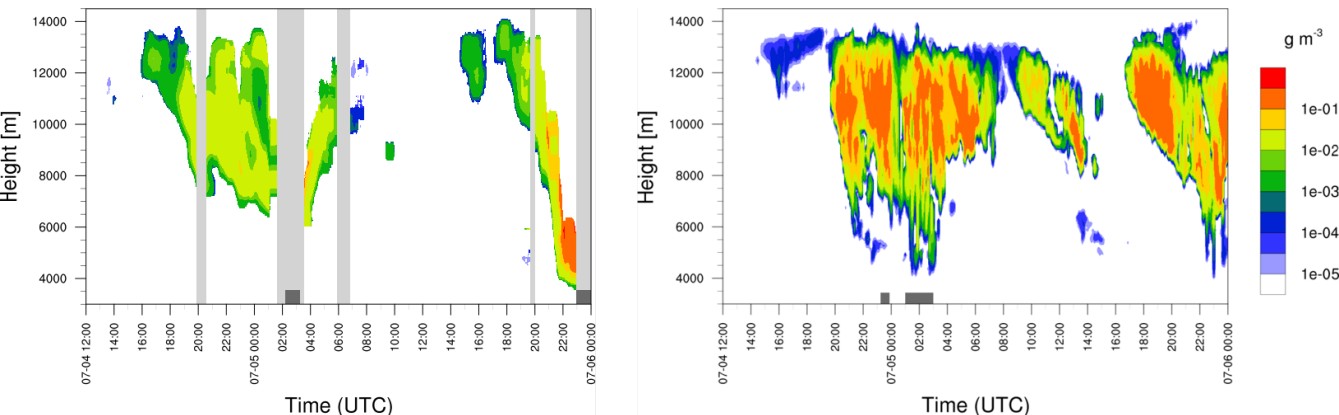

**Figure 4.** Comparison of observed (RAMSES, left) and simulated (ICON-LEM, right) temporal evolution of IWC on 4-5 July 2015. RAMSES IWC was retrieved from measurements of particle depolarization ratio and backscatter coefficient (see the Appendix A1 for more details), light grey-shaded bars indicate measurement breaks for operational (day-night transitions, calibration) and environmental (precipitation) reasons. Dark grey boxes at the bottom of the plots show measured and simulated surface precipitation, respectively.

tions. Instead, the simulation is compared with RAMSES measurements at the Lindenberg Meteorological Observatory (Fig. 4). The juxtaposition shows several features. The measured and simulated cloud top heights match well. The
5 apparent decline in RAMSES cloud top height between 1 and 6 UTC and after 20 UTC on 5 July 2015 is caused by strong signal attenuation and does not reflect the actual cloud vertical extent. The overall temporal development of cloud geometrical thickness during the 2x24-hour ICON-LEM simula-
10 tions agrees well with RAMSES observations over Lindenberg, with the bulk of IWC being between 7 and 13 km in both model and observation. The simulation of precipitation yields mixed results. Simulated precipitation intensity was compared to estimates from attenuated backscatter profiles
from ceilometer observations which were confirmed by rain gauge measurements. While precipitation between 02:15 and 03:15 UTC is well reproduced, ICON-LEM misses the heavy rainfall starting at about 23 UTC on 5 July 2015. Although patches of precipitation can be found in the neighbouring
ICON-LEM grid points, the precipitation intensity is lower than in observations. The inability to simulate heavy precipitation is likely the result of the simulation failing to reproduce the downward movement of the cirrus bottom height to altitudes below 4 km and the accompanying rise in IWC. In
contrast, the short-lived precipitation predicted for approximately 23:30 UTC on 4 July 2015 is locally very confined in the simulation and is not confirmed by observations. Despite the good agreement in the temporal development, the magnitudes of RAMSES IWC and ICON-LEM IWC generally
disagree. Between 20 UTC, 4 July 2015, and 20 UTC, 5 July 2015, the simulation predicts higher IWC values throughout the cirrus core than the RAMSES retrieval, while before and after this period (and below 6 km) the discrepancy is the opposite. This disagreement is unlikely to be caused by the
comparison of a ground-based 1D observation with the sim-

ulation at a single model grid point, since in both observations and model simulations Lindenberg is situated well under the convective anvil, unless the anvil is very inhomogeneous. A noteworthy exception is the evening of 5 July 2015, when RAMSES IWC and ICON-LEM IWC are comparable 40 above 6 km. Differences go either way and can be significant (up to more than one order of magnitude). Clearly, the question arises how to explain this IWC mismatch given that reasonable agreement between ICON-LEM IWC and Cloudnet IWC has been found for 20 June 2013. As can be seen in 45 the following sections, the likely reason is that ICON-LEM simulated the different synoptic situations with varying skill. The 20 June 2013 case was in many aspects a well-simulated day, whereas the predictability of 4-5 July 2015 appeared to be significantly lower and thus ICON-LEM struggled to sim- 50 ulate ice cloud properties realistically. This statement is supported by an evaluation of organizational indices for the 4-5 July case, indicating a lower performance of the diurnal cycle of cloud-top organizational state (Pscheidt et al., 2019). Additionally, a comparison of RAMSES IWP with the satel- 55 lite retrieved IWP product of SPARE-ICE (Fig. A1) shows a good agreement for 4 July 2015, indicating a thinner cirrus cloud over Lindenberg than simulated by ICON-LEM.

### 5.2.2 Comparison to satellite observations

In order to further evaluate the representation of cloud ice, 60 a comparison with the following satellite cloud products has been performed: SEVIRI CiPS, SEVIRI APICS, SEVIRI SatCORPS, SEVIRI CPP, MODIS, and SPARE-ICE (Sect. 4.3-4.8).

Figure 5 and Fig. 6 show observed and modeled values 65 of ICC, IWP and CTH. The shaded yellow-orange area in modeled ICC and CTH represents simulated ICC or CTH calculated for the two different $IWP_{CiPS-sim}$ thresholds (Sect. 4.10). Spaceborne observations (CiPS) of ICC and

CTH are plotted with a continuous black line. As far as IWP is concerned, the spread between modeled tqi and tqf is also represented by a shaded yellow-orange area. The three geostationary MSG/SEVIRI satellite observations of IWP are represented with three different line types (SatCORPS: dotted, APICS: continuous, CPP: dashed) and the spread in observations is represented by a shaded grey area. Since during night only SatCORPS is able to retrieve IWP of thick clouds, only one curve remains and there is no shaded area. Polar orbiting IWP observations are denoted by red symbols (MODIS: circles, SPARE-ICE: triangles).

Figure 5 shows the temporal evolution of the domain averaged ICC as compared to CiPS and IWP compared to the above mentioned data sets over Germany for all three days. Focusing on 20 June 2013 (Fig. 5a), the fairly accurate simulation of the temporal development of ICC is evident. The increase in observed ICC after 11 UTC, connected with the approaching frontal zone and the embedded convection, is well reproduced by the model in terms of timing and amplitude. The underestimation of ICC by ICON-LEM in the morning is related to the failure to resolve an early morning cirrus cloud field. The overall sensitivity of the results to the inclusion of thin cirrus is low on this day, as reflected by the small shaded yellow-orange area.

The analysis of IWP for 20 June 2013 (Fig. 5a) reveals several important aspects. A huge difference (up to a factor of 3) between simulated tqi and tqf (see Sect. 4.10 for the definition of these two variables) is apparent, indicating a substantial amount of graupel and snow (and to a minor extent hail) in the ICON-LEM simulations. Including large ice particles in the calculation of the model tqf results in a strong overestimation compared to observations during the convective phase of the frontal zone (after 12 UTC). However, during this day the amount of SEVIRI pixels inside the cloud where the upper threshold for observable optical thickness and thus IWP are reached amounts to 20 % already at ca. 11 UTC (depending on the single retrievals, see Sect. 4.9 and the single retrievals descriptions). This implies that IWP in this case could be significantly underestimated by the passive retrievals, unless compensation effects occur (Sect. 4.9). Worthwhile mentioning is that in this case both APICS and MODIS, that use different thresholds and have two different spatial resolutions, remain very close to each other shortly after 12 UTC, thus pointing out that the threshold selection does not induce a strong variability in the VIS-NIR retrievals at this stage, maybe due to the still small spatial extension of the convective cell. The modeled total ice amount is biased high even compared to SPARE-ICE retrievals, which are not affected by saturation issues and are generally considered more representative of total as opposed to cloud ice. All observational data sets rather provide IWP values similar to the simulated tqi estimate consisting of small cloud ice particles only. The largest IWP discrepancy between the observations is found during the strong convective phase between 12 UTC and 18 UTC, when the percentage of saturated VIS-

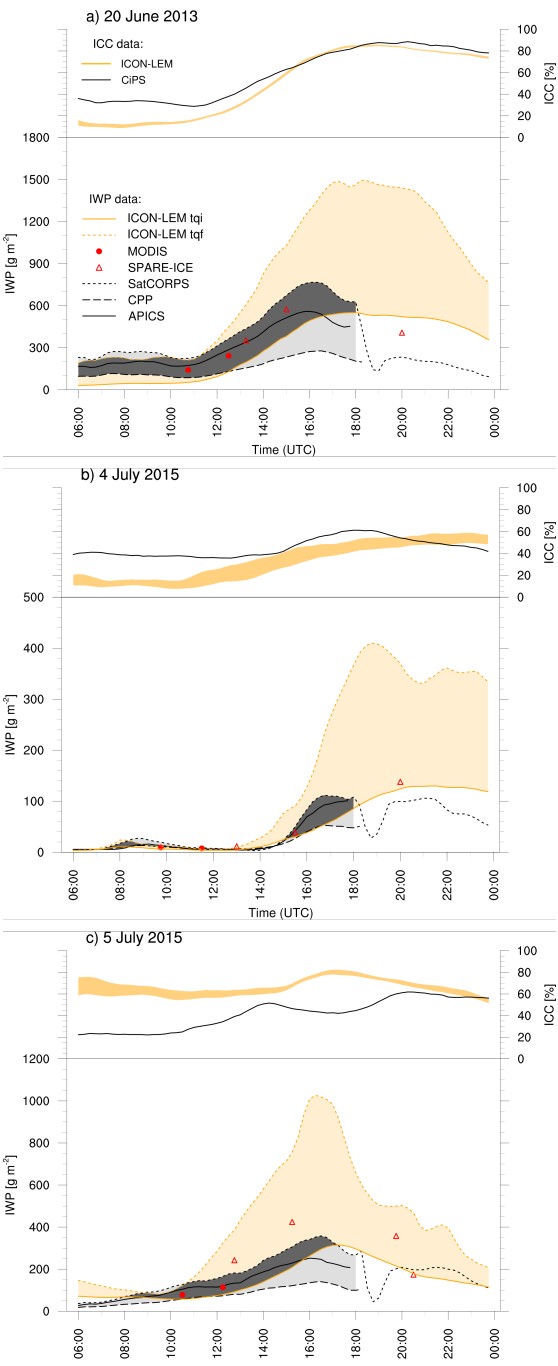

**Figure 5.** Temporal evolution of domain-averaged simulated ice cloud cover (ICC, right axis; top part of each figure) and integrated ice water path (IWP, left axis, bottom part of each figure) and corresponding satellite observations. The simulated range of ICC and IWP is displayed by the orange shaded region, whereas the observed range of IWP by geostationary VIS-NIR retrievals is displayed in light grey. Modeled IWP is separated into two variables differentiating column integrated cloud ice crystals (tqi) with respect to all ice habits (tqf; see sub-section 4.10 for further explanation). The dark grey region shows matching model and observational range. Symbols denote polar orbiting IWP observations (MODIS: circles, SPARE-ICE: triangles).

NIR retrievals is the highest. As discussed in Sect. 4.9, the maximum reported optical thickness and to a lesser degree the maximum reported ice crystal effective radius vary significantly between the different data sets, resulting in a large scatter in domain-mean IWP when the scene is dominated by deep convective clouds. Also note that the SatCORPS and SPARE-ICE retrievals indicate a faster IWP decay, i.e. cloud thinning, after sunset than simulated by the model, while the modeled and observed cloud fractions agree well. The underestimation of tqi before 12 UTC is consistent with the underestimation of ICC in the morning. Please note again that MODIS data is always close to the APICS curve or between the APICS and CPP values. SPARE-ICE IWP is close to the APICS line or between APICS and SatCORPS during day, despite its enhanced sensitivity to larger ice hydrometeors as explained in Sect. 4.9. SatCORPS is almost always larger than the other VIS-NIR retrievals, even in non convective situations (e.g. in the morning hours of 20 June 2013) where different hydrometeors types than cloud ice shouldn't be relevant, thus indicating a slightly different approach to IWP than the other algorithms. During night SPARE-ICE IWP is larger than SatCORPS IWP on this day. In general, CPP seems to retrieve less thick clouds and its increase in IWP after convective initiation at around 11 UTC is also slower.

The analysis for 4 July 2015 (Fig. 5b) shows larger differences with regard to ICC. The area coverage of simulated cirrus cloud fields in the morning is strongly underestimated compared to CiPS. This is due to the outer edge of a front consisting mainly of thin cirrus passing over Central Europe that is not captured by the model but is observed by CiPS thanks to its high sensitivity to thin ice clouds. An increase in ICC after 10 UTC (before convective initiation) is noticeable within the ICON-LEM simulation partly compensating the lack of ICC.

The start of the convective activity in the ICON-LEM simulations ($\sim$ 13 UTC) and observations ($\sim$ 15 UTC) is roughly the same. But convective triggering in the simulations appears to continue well into the night which could not be supported by satellite observations. ICC is comparable with CiPS after the main convective event and consists of a larger cirrus system connected with the convective outflow. The maximum ICC values are similar for both ICON-LEM and CiPS (approx. 60%), but CiPS reaches its maximum ICC at around 18 UTC while ICC from ICON-LEM steadily increases from 10 to 24 UTC. In a simulation that was run for 2 consecutive days we found that the life time of the anvil was significantly overestimated. The width of the shaded area in ICC implies that approximately 10 % of the total ICC consists of clouds with very low optical depths (around 0.05 to 0.14) introducing also a large uncertainty in the determination of simulated cloud top heights depending on the assumed $IWP_{CiPS-sim}$ thresholds (see Fig. 6). In combination with the development of ICC, the IWP strongly increases after initiation of convection around 14 UTC, but reaches both in simulation and observation lower peak values than on 20

June 2013 (Fig. 5a) and 5 July 2015 (Fig. 5c). On this day (4 July 2015) the tendency of IWP in the observations is very steep and resembles the increase in tqf rather than in tqi. However, at 16 UTC the maximum IWP is reached in the observations and its value agrees very well with the model tqi.

The IWP estimates of SPARE-ICE and the SEVIRI retrievals agree well for 4 July 2015. In the morning almost no cloud ice is simulated, despite the fact that ice clouds (with ICC $\approx$ 40 %) are apparent indicating that the cirrus field is optically very thin. The comparison between simulated and observed IWP during the convective phase shows similar results as for 20 June 2013: considering only cloud ice particles and neglecting snow, graupel and hail, tqi agrees well with satellite estimates. Please notice that in this case the SEVIRI retrievals were almost not affected by saturation, with only a few percent of pixels reaching the maximum optical thickness. Overall, the explosive convection triggered around 14 UTC exhibits a much more complicated synoptic situation to be represented by the model, as will be shown in Sect. 6.1, resulting in a poorer matching of observed and modeled IWP than for the 20 June 2013 case.

Satellite estimates are subject to saturation effects (see Sect. 4.9), so that it is advisable to apply an upper threshold to the model results when using them for evaluation. Applying an IWP cut-off threshold of 10,000 g m$^{-2}$ (upper limit of SPARE-ICE) reduces simulated domain averaged tqf at times of peak ice water path by approximately 15-20 % during all three convective events. Applying a saturation threshold to ICON-LEM tqi leads to negligibly changed estimates. Even when using the lowermost cut-off threshold (representing the saturation limit of MODIS) of 3700 g m$^{-2}$ the maximum reduction amounts to 0.2 %. Around 1-3.5 % of the model grid points at times of peak convective activity (20 June 2013: 3.5 %; 4 July 2015: 1 %; 5 July 2015: 2.5 %) display values higher than this threshold. Therefore, restricting the range of simulated IWP values does not alter the finding that ICON-LEM tends to overestimate total IWP.

A comparison between CTHs of ICON-LEM and CiPS has been performed in Fig. 6. Histograms display the frequency of modeled and observed domain-averaged CTHs for each day separately and the width of the lines represents the uncertainty connected with the detection efficiency of the CiPS algorithm (Sect. 4.10). Despite the different synoptic situations for these days, ICON-LEM shows on average the same peak in CTH at approximately 12.5 km for all days. The observed CTH from CiPS is, however, more variable. On 20 June 2013 (top panel in Fig. 6), the model almost perfectly captures the shape of the CTH distribution, but with a constant bias of approximately 1 km. This could partly be a result of CiPS' tendency of underestimating the CTH for unusually high cirrus clouds in mid-latitudes. Validation against CALIOP (Strandgren et al., 2017a, Fig. 10) shows that at German latitudes CiPS retrieves almost bias free CTHs for ice cloud tops located between approx. 8 and 11 km, while it

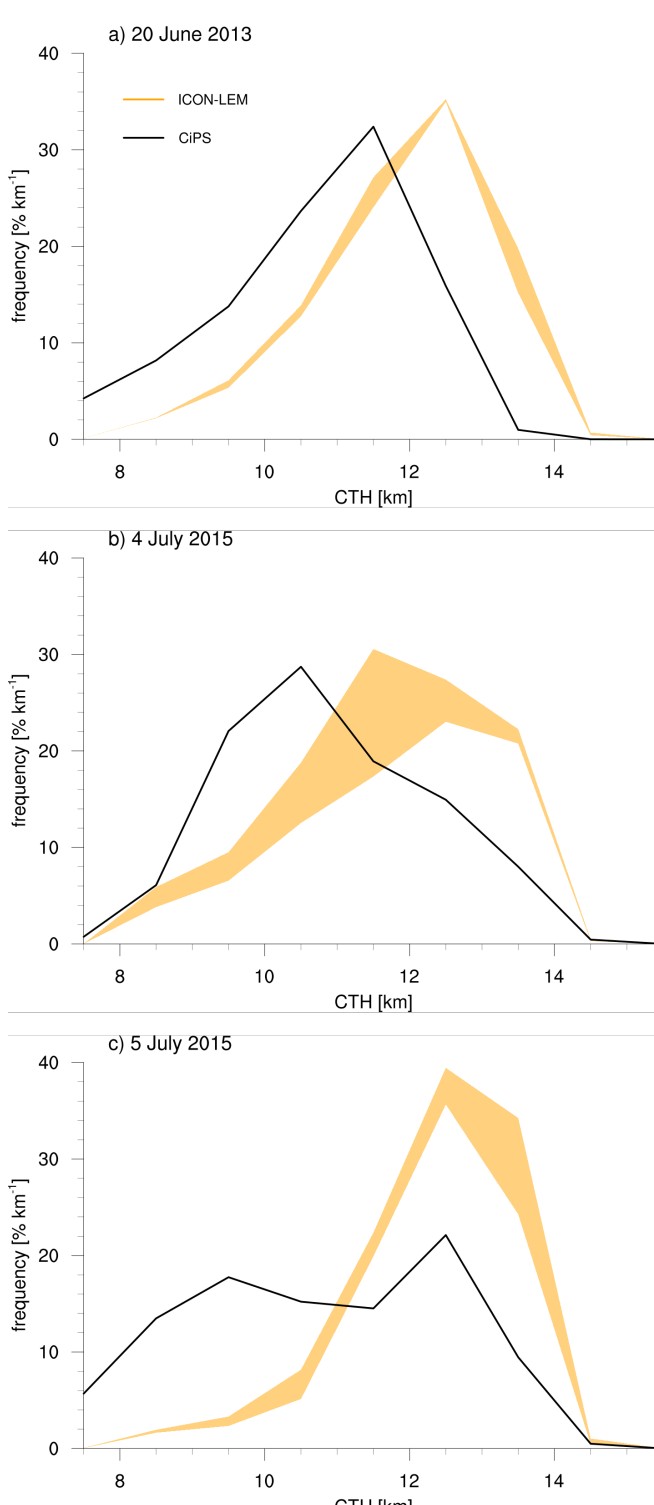

**Figure 6.** PDF of simulated cloud top height for 20 June 2013 (a), 4 July 2015 (b) and 5 July 2015 (c) compared with CiPS. The shaded area shows the sensitivity to two different IWP thresholds ($0.6\,\mathrm{g\,m}^{-2}$ and $3.0\,\mathrm{g\,m}^{-2}$, see Sect. 4.10) considering thin cirrus clouds.

tends to underestimate CTHs that are higher than 11 km and to overestimate CTHs that are lower than 8 km. In particular, CiPS underestimates CTHs in the range 11 to 13 km by approx. 1 km on average for the geographical location analyzed in this paper, which is in line with the difference between observation and model. Nevertheless, the occurrence of lower cloud top heights of up to 10 or 11 km are likely underestimated in the simulation. On 4 July 2015, the modelled CTH again peaks at approx. 1 km higher altitudes than in the observations. Furthermore, the distribution of the modelled CTH is skewed towards higher CTH, whereas the distribution of observed CTH is skewed towards lower CTH. Those differences do not merely result from the fact that the early morning cirrus cover was not reproduced by ICON-LEM. Instead we see that additionally low ice clouds are missed by the model later in the day. CiPS indicates that CTHs are lower as one moves away from the convective core, whereas ICON-LEM simulates more homogeneous cloud top heights over the whole cirrus shield (Appendix A2). The modeled cloud top heights therefore result in a more distinct CTH peak displayed by the histograms. A rather uniform distribution of observed CTHs is apparent for 5 July 2015 which is not reproduced by ICON-LEM. The large probability of high CTHs and the corresponding lower probability of lower CTHs in the simulation may partly be due to the model predicting an excessively long-lived of the outflow cirrus that maintained high CTH. Again, ICON-LEM seems to miss the decrease in cloud top heights at the edges of the convective cloud field. For all days, the maximum simulated CTH agrees well with the observed maximum height of 14 km, which is important in order to capture the effect of the cloud field on longwave radiation.

## 5.3 Statistics of several convective days

In order to provide an analysis of ICON-LEM performance over a broader range of convective situations, we have collected eight convective days in the time period 2013-2016 (Table 1). This selection, which also includes the three days evaluated in the previous sections, encompasses different kinds of meteorological conditions, from convection embedded in fronts to scattered convection. For all these days we evaluate statistics of CTH, ICC, and IWP.

The simulated CTH distribution shows good agreement with the observed one (Fig. 7a). As mentioned above, the slight rightward shift of the simulated CTHs to higher values compared to observations is partly explained by the known negative bias of CiPS underestimating unusually high CTHs at mid-latitudes (see Sect. 5.2.2). The model, however, underestimates the frequency of clouds with CTHs at the lower end of the distribution between 8 and 10 km. This is partly caused by the overestimation of the height of anvil edges, which is present in all convective simulations and is particularly strong in the convective situations on 4-5 July 2015. As shown in Appendix A2, the CiPS observations show that

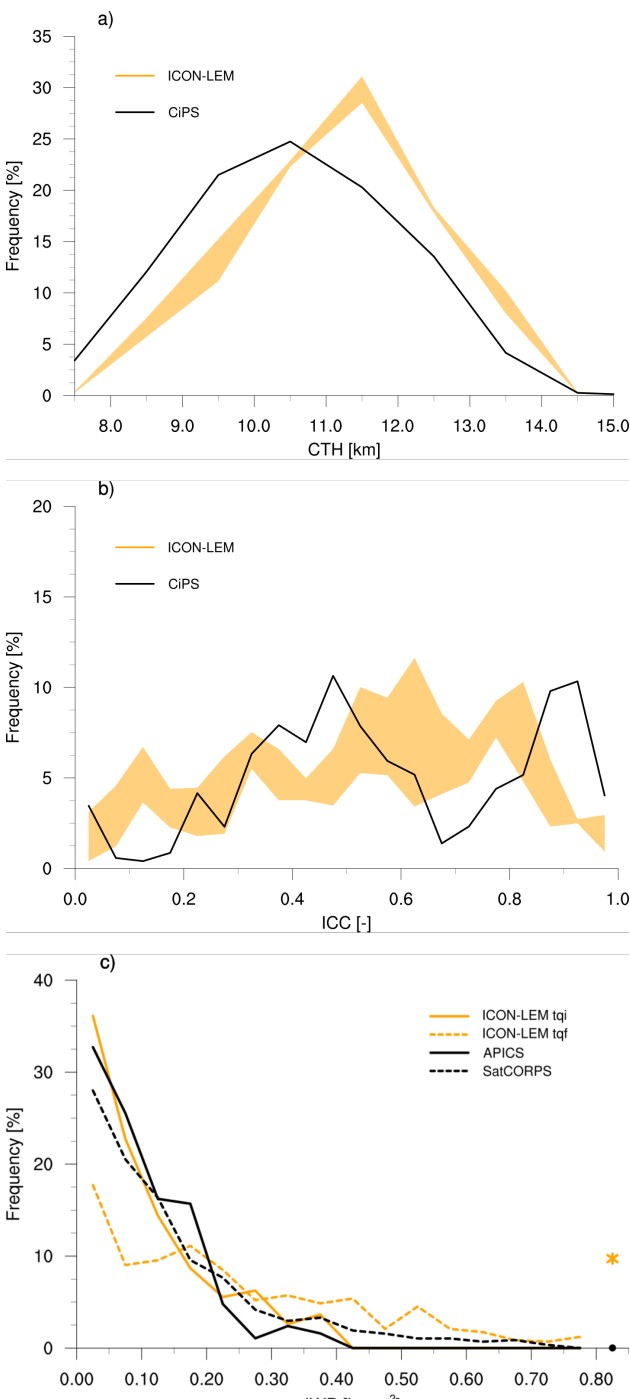

**Figure 7.** Histograms of cloud top height CTH (a), domain average ice cloud cover ICC (b) and IWP (with bin size of 0.05) (c) for all simulated convective days listed in Table 1. The observational CiPS data set is used as comparison for CTH and ICC while APICS and SatCORPS are used for IWP. Simulated and observed IWP data are restricted to daytime values between 06:00 and 17:30 UTC due to the limitation of APICS to sunlit hours. In (c) the orange star indicates accumulated frequencies with simulated tqf larger than $0.8\,\mathrm{kg\,m^{-2}}$ and the black dot shows the accumulated frequency of ICON-LEM tqi as well as of both observational IWP estimates.

the thunderstorm cloud has highest CTHs in and around the convective core and that CTH decreases towards the cloud edge (the blue colours in Fig. A2, top). The model, bottom panel in Fig. A2, also shows CTH peaks in the inner part of the cloud, but it lacks the realistic simulation of the CTH distribution towards the cloud edges.

The interpretation of the ICC and IWP histograms is more difficult, because our ensemble of simulations consists of a few large scale convective events partly connected with frontal systems and a few cases of scattered small scale convection. Therefore, the convective activity does not always lead to the largest ICC and IWP when averaged over the simulation domain. The histogram of ICC (Fig. 7b) shows a relatively flat distribution with maxima in observed ICC around 50% and 90% cloud coverage. In the simulations the highest probability is for ICC between 50% and 80%, but a large part of those ice clouds are optically thin. The differences in the observed and simulated ICC histogram may have different causes. They could be related to an underestimation of the convective cell extension, even though the opposite seems to be true for the 4-5 July case, to an underestimation of ice clouds originating from other meteorological systems that remain unresolved in ICON (see Sect. 5.2, discussion of ICC for the morning of 20 June 2013), to spatial shifts of the convective spots that partly evolve outside the ICON domain, or to errors stemming from the initialization.

Concerning the IWP histogram (Fig. 7c), the ICON-LEM tqi generally follows the observations from SatCORPS and APICS well. Maximum simulated tqi and APICS values are $\approx 400\,\mathrm{g\,m^{-2}}$ while SatCORPS retrieves IWP values of up to $\approx 700\,\mathrm{g\,m^{-2}}$. For IWP between $250\,\mathrm{g\,m^{-2}}$ and $700\,\mathrm{g\,m^{-2}}$ ICON-LEM tqf is well aligned with SatCORPS, whereas for smaller IWP values the SatCORPS frequencies lie between the two model curves. Nevertheless, a significant amount ($\approx 10\,\%$) in IWP frequency for tqf lies above $800\,\mathrm{g\,m^{-2}}$ indicated by the star at the end of the distribution, which is not apparent in the observations or ICON-LEM tqi. Maximum domain averaged (daytime) values are $1400\,\mathrm{g\,m^{-2}}$ for tqf, which were found on 20 June 2013, the day with the most extreme IWP values (see Fig. 5). Compared to SatCORPS, APICS shows a higher occurrence of thin ice clouds thanks to the significantly higher sensitivity of CiPS used for ice cloud detection in this retrieval (Sects. 4.4 and 4.3). Those thin cirrus clouds are largely missing from the model (and also from the other satellite retrievals), as discussed for the 4 July 2015 (see Sect. 5.2). At the other end of the IWP distribution, APICS does not provide as high domain averages as SatCORPS, since the maximum retrieved APICS IWP values are lower (see Sects. 4.4, 4.5 and 4.9). APICS generally follows the ICON-LEM tqi curve well, except for the range $150$–$200\,\mathrm{g\,m^{-2}}$, where APICS and ICON-LEM tqf are better aligned. While the distribution of ice in the model is generally similar to satellite observations, the distinction between tqi and tqf can be considerably different between model and satellite retrievals, and also between the various retrieval al-

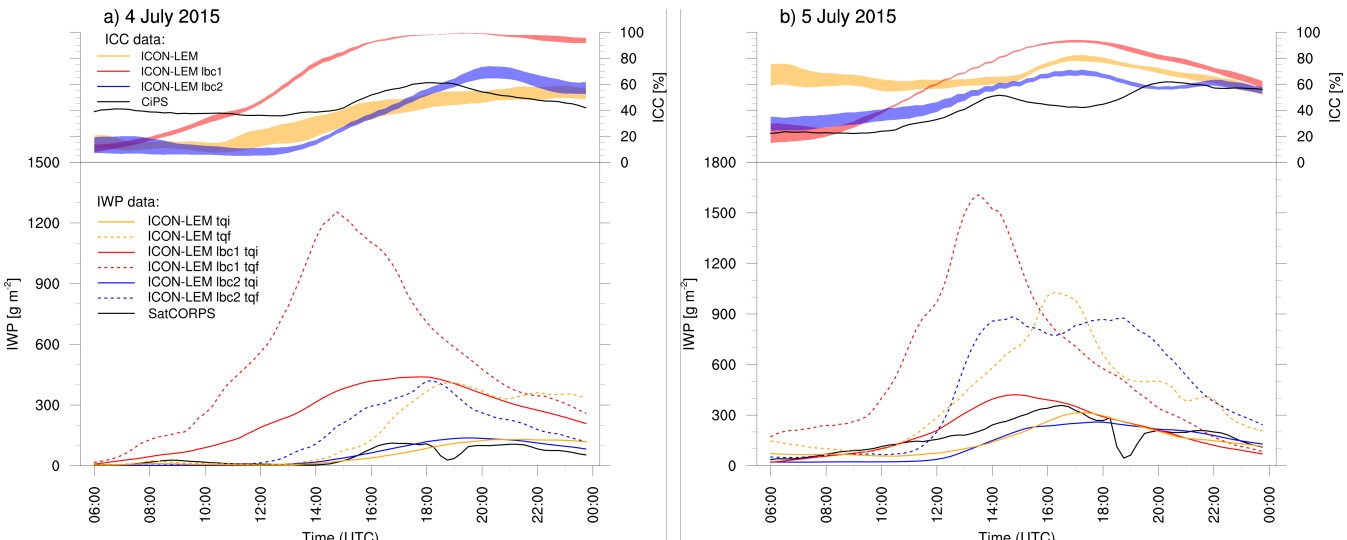

**Figure 8.** Similar to Fig. 5, but displaying the change in the temporal evolution of ICC and IWP when varying the initial conditions for 4 July 2015 (left) and 5 July 2015 (right). Only SatCORPS observations are shown as a reference. The yellow, red and blue lines correspond to simulations forced with lateral and initial conditions from COSMO, ICON-NWP and IFS respectively.

gorithms (Sect. 4.9), due to the different sensor sensitivities and assumptions made on partitioning the total ice into the various ice habits.

## 6 Sensitivity studies

Here we investigate the possible causes of model deficiencies noted in the simulations of the convective situations on 4-5 July 2015, e.g. the excessive anvil life time. We selected this case because of the large convective instability related to large CAPE values and expect that small differences in boundary conditions and/or model physics perturb the simulations enough to shed light on these deficiencies. To explore potential error sources we ran sensitivity experiments with modified model physics, in particular modified cloud microphysics (Sect. 6.2), and with changing initial and boundary data used to drive the model (Sect. 6.1), giving a measure for the predictability of the synoptic situation.

Note that the sensitivity studies were performed at 625 m resolution with no further nesting in order to save computing time and storage space - as opposed to 150 m resolution for the simulations discussed above. As Stevens et al. (2020) pointed out, the improvement going from 625 m to 150 m is modest, so we expect the results of our sensitivity study to carry over to the higher resolution domain. A comparison of the two control simulations at 625 m and 150 m resolution confirmed this; for example, cloud water path (tqc) and tqi only changed by 1.5% and 6.0%, respectively.

### 6.1 Sensitivity to initial and boundary conditions

For 4-5 July 2015 additional simulations were performed using different initial and lateral boundary conditions. Instead of using initial and boundary data from the COSMO-DE analysis fields (in the following referred to as "default simulation"), data from ICON-NWP (lbc1) and the IFS (lbc2) have been used (see Table 2). The sensitivity simulations using IFS (cycle 41r1) and ICON-NWP data were analyzed regarding the evolution of IWP, ICC and the distribution of CTHs and compared to the default simulation and observations (Fig. 8 and Fig. 9).

In all three simulations strong convective events are located over northern Germany on 4 July 2015. However, both the timing and the amplitude of the increase in IWP and ICC (Fig. 8a) appear to be very sensitive to the initial and boundary data. In A3 we analyze initial and boundary data from the three driving models that lead to those differences. Using the ICON-NWP data for initialization, convective activity starts too early and is too vigorous with overestimated CTHs (Fig. 9). This appears to be caused by the a moist bias in the boundary layer in the ICON-NWP analysis (see Fig. A3c) and occurs despite the stabilizing effect of the small temperature warm bias in the middle troposphere (see Fig. A3a in Appendix A3). Using COSMO-DE or IFS data for the initialization and boundary conditions ICON-LEM captures the temporal evolution of the IWP over Germany well. The SatCORPS IWP estimate agrees well with simulated tqi in the default and lbc2 simulations, whereas in the lbc1 simulation tqi is much larger than observed. The decrease of tqi at the end of the day is not captured in the default simulation. The evolution of ICC is slightly less successfully simulated. ICC

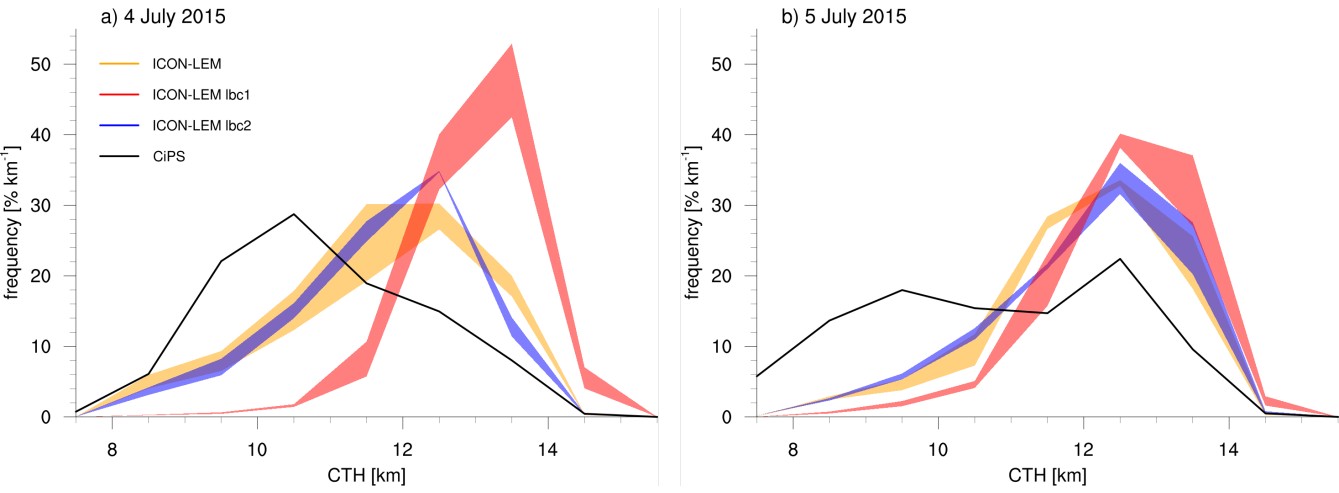

**Figure 9.** Similar to Fig. 6, but displaying the change in CTH when varying the initial conditions for 4 July 2015 (left) and 5 July 2015 (right).

is underestimated in the morning and decreases only slightly (lbc1 and lbc2) or fails to decrease completely (default) during the night, indicating that the cirrus field connected with the convective outflow remains too large for many hours after the main convective event. In the ICON-LEM default simulation for 4 July 2015, tqi remains constant and ICC continues to increase through the night. As pointed out before, this continued increase in the modeled cirrus shield appears to be caused by the numerous small convective events simulated in the vicinity of the convective cirrus shield throughout the afternoon and night, which are in contrast with the single big convective event observed in the afternoon. CTHs in those two simulations are lower than in the ICON-NWP forced simulation, but the fraction of clouds reaching 13 km is significantly too high when compared to observations (Fig. 9a). For all three simulations tqf is significantly higher than tqi. The difference is particularly large at the time of convection and several hours afterwards pointing to a large number of larger hydrometeors. Whereas the difference between tqi and tqf strongly decreases at night in the lbc1 and lbc2 simulations, this is not the case for our default simulation indicating a continuing large optical depth of the ICC resulting from the convective event.

The spread in the simulations for 5 July 2015 (Fig. 8a) is slightly smaller than for the previous day despite similar problems with the initial and boundary conditions coming from the three data assimilation systems (Fig. 8 Appendix A3). This is likely due to the 5th July being dominated by large scale convective forcing along a frontal system. The start of convective activity in the ICON-LEM lbc1 run is slightly too early, which is likely connected with a premature transition of the frontal system in the morning in ICON-NWP. The default simulation starts with an ICC significantly too large and a small overestimation of IWP, both associated with the excessive life time of the convective cirrus shield.

This suggests that COSMO-DE which supplies the initial and boundary data for the default simulation also overestimated the life time of the ICC resulting from the large convective event of the previous day, a model error that is not shared by the IFS or ICON analysis forced runs (Fig. A3e). Simulated tqi agree reasonably well with SatCORPS data with the lbc1 simulation significantly overestimating tqi in the early afternoon. Simulated CTHs (Fig. 9b) agree better with observations than for the previous day and show convection reaching up to 13 km.

In general, CTH distributions do not vary strongly with initial and boundary data for these two days, except for the overestimation of the CTH on 4 July 2015 when using ICON-NWP data. Furthermore, simulated CTHs underestimate the frequency of lower cirrus clouds on both days (Fig. 9). While the observed distribution of CTH appears wide or even bimodal, the model prefers single-peaked distributions centered on high CTH between 11 and 14 km, capturing little of the lower level cirrus fields that CiPS detects between 8 and 10 km. The absence of lower CTHs is caused by the overestimation of CTH in clouds not directly connected with the convective systems and also by the overestimation of CTH at the edges of the convective cirrus shield (see Appendix A2).

## 6.2 Sensitivity to microphysics

To investigate the representation of cloud microphysical processes as a possible cause of model deficiencies, we have performed 16 sensitivity studies with different microphysical assumptions (Tab. A1). The sensitivity experiments have been performed for 5 July 2015 with a grid spacing of 625 m. The microphysical control run has the same configuration as the default simulation in Tab. 2 without adding further nested domains.

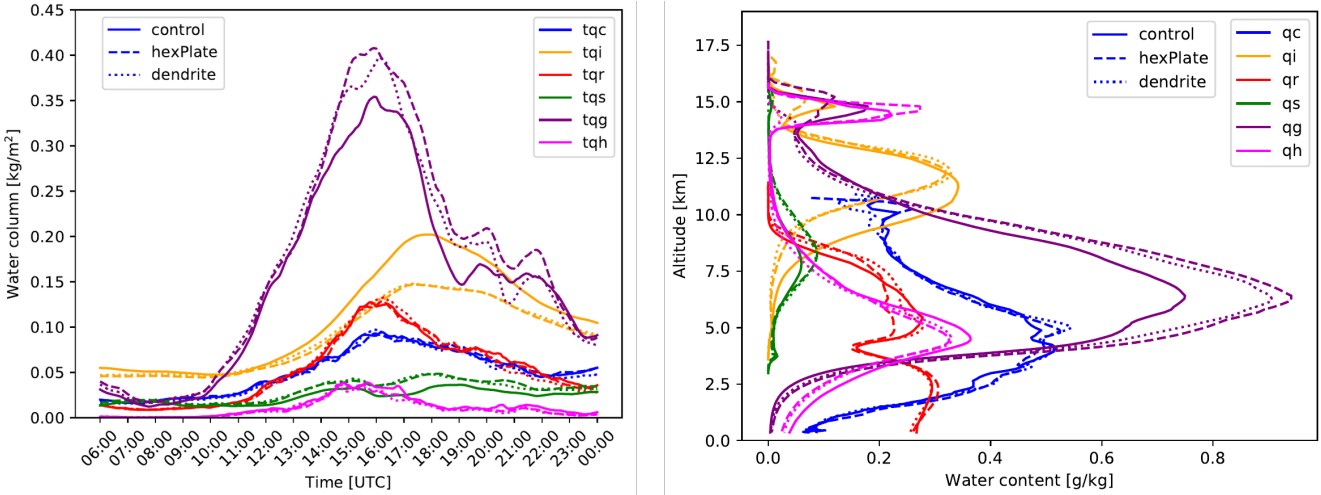

**Figure 10.** Temporal evolution of the atmospheric liquid and ice water paths on 5 July 2015 (left) and in-cloud water content profiles at 16 UTC (right) with three (control, 'hexPlate, 'dendrite') cloud ice geometries.

Here we discuss these experiments, which all lead to a reduction of IWP; recall that an over-pronounced anvil cloud has previously been identified as a likely model bias. A short description of the experiment setups and their outcome is given in Appendix A4. We concentrate on three experiments in particular. The experiments 'hexPlate', 'dendrite', and 'stickLFOhigh' (Tab. 3 and experiments 3, 4, and 10 in Tab. A1) replace the original mass-size and velocity-size relations for cloud ice by a different particle geometry. The corresponding relations in the control run are for irregular crystals derived from in-situ measurements collected during CRYSTAL-FACE in Florida 2002 (A. Heymsfield, pers. comm.). These irregular crystals have rather high terminal fall velocity more typical of column-like particles. This has been replaced by a plate-like geometry in experiment 'hexPlate' and a dendrite-like geometry in experiment 'dendrite'. Both of these crystal geometries have rather low fall speeds, but they differ in the exponent of the mass-size relation (see Tab. 3), leading to the dendrite-like geometry growing more quickly in maximum dimension than the plate-like crystals. Both experiments result in a significant decrease in cloud ice water path (tqi, Tab. A2) of 18 % and 16 %, respectively. Figure 10 displays the actual time series of the condensate path and the vertical profiles of the in-cloud water content for each water species. This shows clearly that experiments 'hexPlate' and 'dendrite' lead to a decrease of tqi during the day when deep convection develops.

The decrease of tqi corresponds to an increase in the amount of graupel. Note that the graupel category should be interpreted more broadly as partially rimed ice and graupel for the SB scheme. This shift is also reflected in the vertical profiles which clearly show a reduced vertical extent of the cloud ice layer for the 'hexPlate' and 'dendrite' experiments, which is easily explained by the reduced sedimentation ve-

locity. The increase in graupel is most probably caused by the increased collection of cloud ice by graupel due to the increased velocity difference between the two categories and, hence, an increased collection kernel. This behaviour differs from the case of isolated cirrus or anvil clouds for which an increased sedimentation velocity leads to a faster fall out of ice into the drier layers below and, hence, a faster dissipation of the ice cloud and consequently a reduced tqi. For the studied mature mesoscale convective system (MCS) our simulations show the opposite behavior, because of the presence of deep condensate layers with snow and graupel below the cloud ice layer. Unfortunately, the experiments 'hexPlate' and 'dendrite' were unable to significantly reduce the areal extent of the anvil clouds and, hence, did not improve the performance of the ICON-LEM model in that regard. In fact, the slower falling cloud ice particles lead to an increase in ICC and CTH, in disagreement with the CiPS satellite retrievals (Fig. 12). Additionally, latent heat release or cloud dynamics did not change significantly. In order to investigate this in more detail, a cloud tracking algorithm could unveil new insights in the life cycle of individual convective cells, which is beyond this scope.

The strongest decrease in the ice water path tqi is shown by the experiment 'stickLFOhigh', featuring a significantly increased sticking efficiency between ice, snow, and graupel. An increase of the sticking efficiency trivially leads to increased collection rates and, hence, to the faster formation of large precipitation-sized particles, which in turn enhances the depletion of cloud ice by faster conversion to graupel. This is clearly visible in the time series and the vertical profiles shown in Fig. 11. The graupel content in mid-levels, however, is actually decreasing for 'stickLFOhigh', which can be explained by the formation of larger and therefore faster falling graupel particles. Compared to the satellite observa-

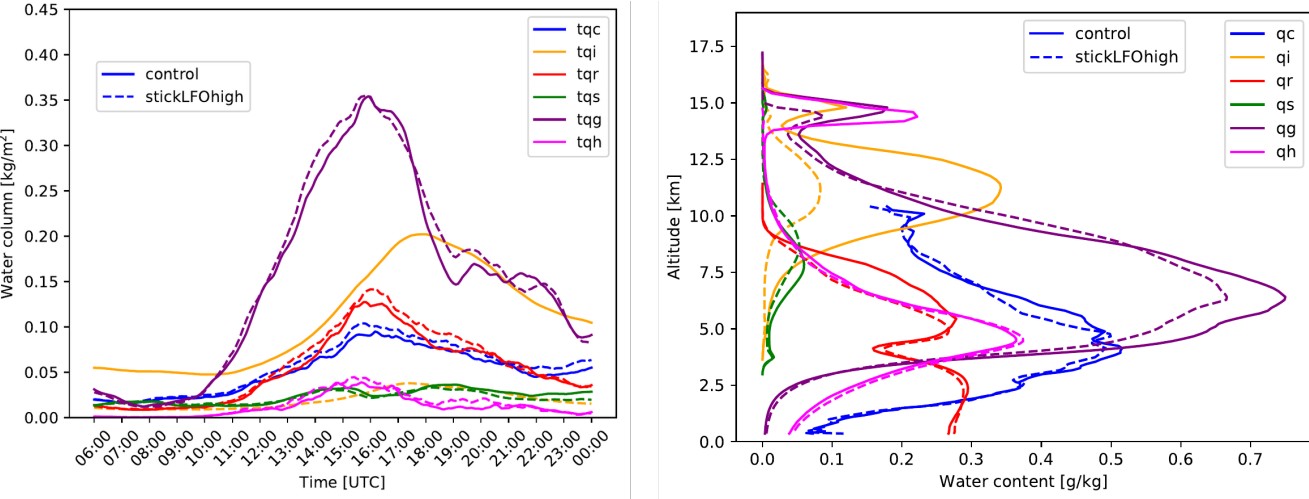

**Figure 11.** Similar to Fig. 10, but for high sticking efficiency ('stickLFOhigh').

tions of cloud top height and ICC, there is no significant improvement, though. The change in sticking efficiency affects mostly the vertical structure of the MCS and less so its horizontal extent. Overall, the 'stickLFOhigh' (Table 3) simulations produced inconclusive results. We also note that the used sticking efficiencies are rather high in light of more recent laboratory measurements (Connolly et al., 2012).

## 7 Discussion and conclusions

A qualitative and quantitative evaluation of summer convective events in large-eddy simulations over Germany has been performed. ICON, as a cutting-edge model resolving deep moist convection with an applied resolution of $O(100\,\mathrm{m})$, gives unprecedented insights into clouds and precipitation for the next generation of NWP models. We examined different cases of summertime convective situations with regard to the timing and strength of the convective transport of water into the upper troposphere as well as the horizontal and vertical extent and the evolution of the resulting convective anvil. Furthermore, we studied the sensitivity of the simulations to initial conditions and microphysics, in order to investigate the uncertainty and predictability of simulated convection. For verification we used observed estimates of cloud top height (CTH), ice cloud cover (ICC) and a variety of ice water content / ice water path (IWC/IWP) products from geostationary and polar orbiting satellite sensors exploiting different data and retrieval approaches (optical thermal, optical VIS-NIR, microwaves) as well as ground-based instruments.

Several different convective situations were considered, connected with either scattered, dynamically forced large-scale or frontal convection. Overall, the model evaluation with the above suite of satellite and ground-based observations shows that the convective situations have been mostly well reproduced concerning the spatial and temporal cloud structure. This is consistent with the work of Stevens et al. (2020) who showed that cloud structure and diurnal variability is improved in high resolution ICON simulations relative to coarser resolution models. The convective event of 4 July 2015 extending into 5 July 2015 proved to be the most difficult to reproduce. We focused our evaluation effort on this large-scale convective event and the subsequent passage of the band of frontal convection and additionally a contrasting very good representation of the large-scale frontal convection on 20 June 2013.

The timing of the start of convective activity on those days, expressed in the nearly simultaneous increase in ICC and IWP, is well captured by ICON-LEM. We use the CiPS algorithm, based on the thermal SEVIRI channels, which is optimally suited to describe the spatial extent and cloud top height of the anvil and their temporal evolution. Comparison with the simulations indicates an overall realistic structure of cloud anvils in terms of CTH and coverage. The simulations even agree well with ground-based observations at particular instrument sites (RAMSES and Cloudnet). But the life time of the cloud systems originating from the convective outflow are shown to be too long in particular in terms of ICC. The evaluation of IWP with observations proved to be difficult due to the large uncertainty in observed IWP values. Using a number of different VIS-NIR satellite retrievals and a retrieval using also microwave data allowed us to characterize the spread of observed IWP estimates that encompasses the ICON-LEM simulated cloud ice water path (tqi). Model and observations agree on the relative strength of the convective water transport into the upper troposphere in the three synoptic cases, with the 20 June 2013 being the case with the largest increase in IWP and 4 July 2015 the one with the smallest increase in IWP. On all three days, the model integrated cloud ice tqi agrees well in magnitude and temporal evolution with the VIS-NIR retrievals although in many

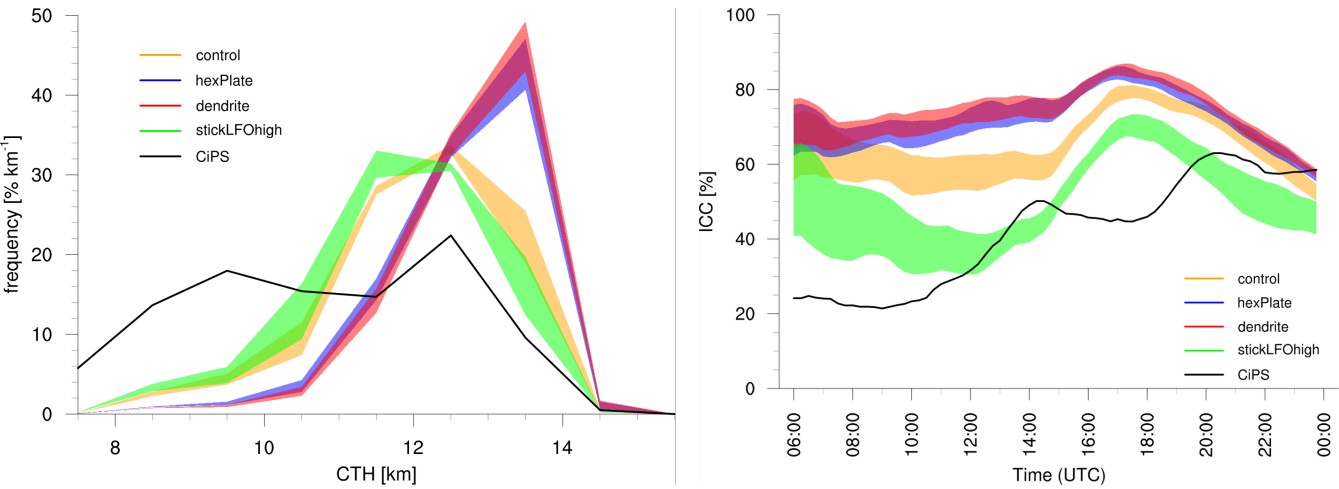

**Figure 12.** PDF of domain averaged CTH (left) and the temporal evolution of ICC (right) when prescribing different ice crystal habits for 5 July 2015.

cases tqi is slightly underestimated by ICON-LEM. Furthermore, in all cases frozen water path (tqf), which is the sum of all ice hydrometeors, exceeds the simulated cloud ice water path and the observed IWP by a large degree as soon as convection is triggered.

Evaluating our ensemble of 8 simulated days regarding CTH, ICC and IWP we find the PDFs of the cloud variables to be reasonably well simulated by ICON-LEM. Whereas CTH is relatively well simulated regarding its variability and its estimate for clouds of convective origin, the evaluation of ICC is challenging since it is very sensitive to the existence of optically very thin ice clouds. The horizontal structure of the CTH of convective anvils appears to be too homogeneous in the simulations and anvil cloud edges are too high (see Appendix A2), which likely hints at deficiencies in the microphysical scheme or cloud-radiation interactions (Gasparini et al., 2019). ICON simulations exhibit a higher probability of very large tqf values than observations. Since observations vary in their sensitivity to different ice habits and cannot detect all ice, a certain overestimation of tqf in the model relative to observations would be expected. However, the model estimate of tqf is in extreme cases, such as 4-5 July 2015, larger than all observed IWPs by a factor of 3 or 4. Therefore the question arises whether ICON can be said to overestimate tqf.

Current state-of-the-art satellite retrievals provide a rather weak constraint on bulk ice mass in the atmosphere. Satellite retrievals employing different remote sensing methods, e.g. involving active and passive instruments, span a large range of IWP estimates. By using remote sensing data in the microwave spectral region, SPARE-ICE is also sensitive to ice hydrometeors other than cloud ice whereas the VIS-NIR retrievals (SEVIRI and MODIS) are not. The VIS-NIR retrievals alone span quite a broad range of IWP that does not appear to be tied to differences in sensitivity to hydrometeors.

Furthermore, when comparing only estimates based on SEVIRI (APICS, SatCORPS and CPP) the spread of retrieved IWP is still significant, up to a factor of 2–3 being typical, due to differences in inherent assumptions. While in many situations SPARE-ICE is close to APICS and/or SatCORPS, particularly in convective situations, it often exceeds all other retrievals. Nevertheless, SPARE-ICE is likely to underestimate tqf partly due to the presence of small ice crystals in convective clouds that cannot be reliably accounted for. The sensitivity of the existing passive and active methods to the different ice habits (small cloud ice versus large precipitating ice) is poorly quantified, complicating the interpretation of the reported IWP values.

What emerges from our model-satellite comparison with confidence is that the simulated tqi is within the current, relatively wide, range of satellite estimates. The model tqf, however, is biased high even compared to satellite estimates based on active radar/lidar retrievals (SPARE-ICE), implying an overestimation of elevated graupel. Measuring the degree of riming would be key to constrain graupel estimates. Recent developments using a video disdrometer (Praz et al., 2017) or vertically pointing radar (Kneifel and Moisseev, 2020; Ori et al., 2020) shed some light on this issue.

Evaluating the ICON-LEM simulations in detail against observations in terms of biases in ice clouds and anvil evolution allows us to go one step further and examine the uncertainty of the associated forecasts at hectometer resolution. Given recent work (see introduction) that points to moist processes and initial conditions and large-scale weather as key players in the predictability of convection as well as larger scale weather phenomena we aimed at exploring those sensitivities on cases specifically selected as potentially most unpredictable (high CAPE, yet low large-scale advection).

For the investigation of uncertainty we selected the explosive convective event over Germany of 4-5 July 2015 for

which the model struggled to simulate the evolution of convection realistically. Looking at high cloud properties in the three sensitivity experiments with COSMO, ICON and IFS initial and boundary conditions we found impact on convective triggering, strength and to a lesser degree on the life time of the convective outflow. The sensitivity in terms of ICC and IWP is of similar order of magnitude as the diurnal cycle. Note, that the variability is larger for the more locally forced 4 July 2015 and smaller for 5 July 2015 which was embedded in a front, pointing to the importance of convective instability.

Second, we investigated the sensitivity to microphysics as it represents a large part of the non-linearities and uncertainty in the model physics. Given a tendency of over-prediction of cloud ice in ICON-LEM, we selected modifications focused on the hydrometeor geometry aiming to reduce cloud ice. It is striking to note that these substantial physics changes result in a large reduction in cloud ice (up to a factor of 5) and smaller changes to cloud top height, but the critical timing of convection including the diurnal cycle, in contrast, changed little. The considered changes in the microphysical parametrization did not reduce the water path of the other frozen hydrometeors either or shorten the life time of the convective outflow cloud field.

In summary, the work we present demonstrates the usability of a $O(100\,\mathrm{m})$ resolution model for forecasting studies or parameterization development of convection including anvil evolution and its uncertainty. Given the fact that a major source of non-linearity in cloud-resolving models originates from cloud physics, the surprising result of our case study of 4-5 July 2015 was the relatively small impact of microphysics in the uncertainty of convective development. We therefore recommend future work to focus on a wider set of cases of locally forced continental summer convective days. Another direction of research to strengthen the understanding in the interplay of large-scale forcing and local physics in the uncertainty of the prediction of continental convection would be to investigate other parts of the description of clouds in models relating to the liquid phase and including lateral mixing in convective cores at sub-grid scales. A statistical intercomparison using multi-site Cloudnet information (following the study of Illingworth et al. (2007)) would allow a more comprehensive evaluation for future hecto-scale NWP models, which has only been performed over single locations so far (Nomokonova et al., 2019; Schemann and Ebell, 2020). The current work highlights the existing limits in using observations to evaluate high ice clouds from $O(100\,\mathrm{m})$ forecast models, which originate from both data and algorithms. The arrival of the new spaceborne radar/lidar system EarthCare in 2022 will provide a driving force in both aspects. This will be followed by the Ice Cloud Imager (ICI) in 2023 on EUMETSAT's second generation polar system, giving significantly tighter observational constraints by exploiting sub-millimeter wavelengths and promising a much reduced (50%) uncertainty in IWP retrievals (Eriksson et al., 2020).

# Appendix A

## A1   RAMSES

RAMSES is a spectrometric water Raman lidar which allows to measure water in all of its three phases. However, because of the extremely weak inelastic scattering by clouds, the condensed phases can only be obtained directly under favorable conditions. To widen the range of applicability, the RAMSES data set of cloud water content (CWC) measurements was searched for a proxy variable that would be easier to measure than CWC directly but would still provide reasonable estimates of CWC at all times. It was found that in the case of cirrus clouds the cross-polarized backscatter coefficient (BSCs) serves this purpose, and an analytic expression for deriving IWC profiles and, by extension, IWP from BSCs and atmospheric temperature was developed [Reichardt; manuscript in preparation]. To validate the RAMSES IWC retrieval technique, a comparative study was conducted in which RAMSES IWP was contrasted with IWP results retrieved from satellite-borne radiometers (CiPS, SPARE-ICE). First results have been presented by Strandgren (2018). Generally, good agreement is found when the observed cirrus system can be assumed to be ergodic. As an example, Fig. A1 highlights the comparison between the RAMSES and the satellite observations on 4 July 2015. Before 19 UTC when the cirrus was optically thin, RAMSES and CiPS IWP values coincide. Later on, as was expected, CiPS IWP falls behind because cirrus optical depth increases to values too high for the CiPS algorithm to be applicable (Sect. 4.3. The earlier SPARE-ICE IWP value (around 19:20 UTC) is much smaller than both RAMSES and CiPS IWP. Possibly, the cloud volumes observed under slant angle (SPARE-ICE) or vertically (RAMSES) differ too much so that the requirement of ergodicity is not met in this case. In contrast, SPARE-ICE IWP at 20 UTC is in excellent agreement with RAMSES IWP retrieved shortly before.

## A2   Underestimation of the probability of low cloud top heights

The analysis in Sect. 5.2.2 shows that the probability of low (below 11 km) CTHs is underestimated in the simulations (see Fig. 6b and c). To elucidate the causes, a snapshot of observed CTHs is compared with the default simulation in Fig. A2. The anvil over northeastern Germany is clearly visible in the evening of 4 July 2015. Whereas the observations show a systematic decrease of convective anvil height towards cloud edges, the simulations lack such spatial gradients in CTH. This model deficiency can be seen on most convective days and is the main reason for the underestimation of low CTHs in the simulations. The effect is strongest on

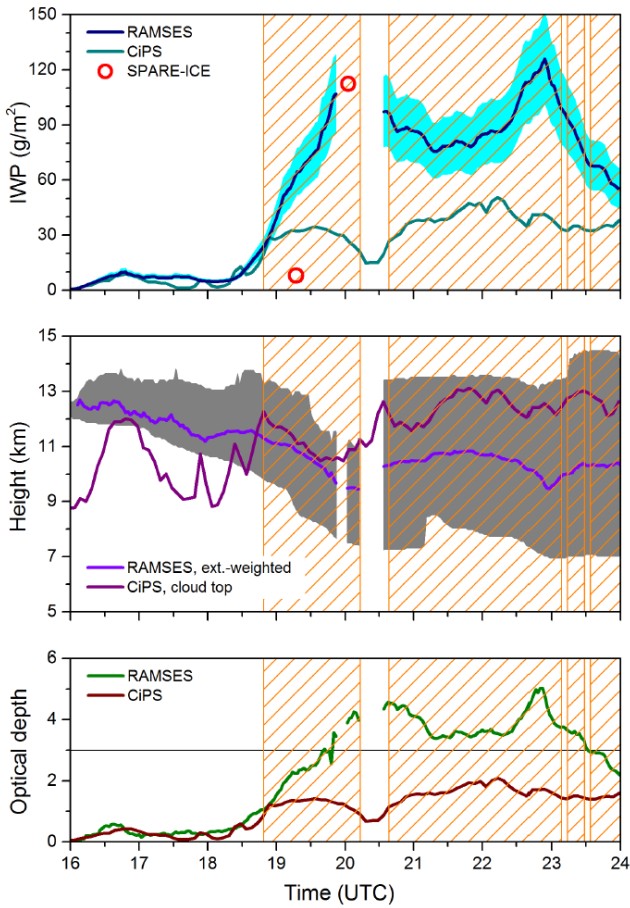

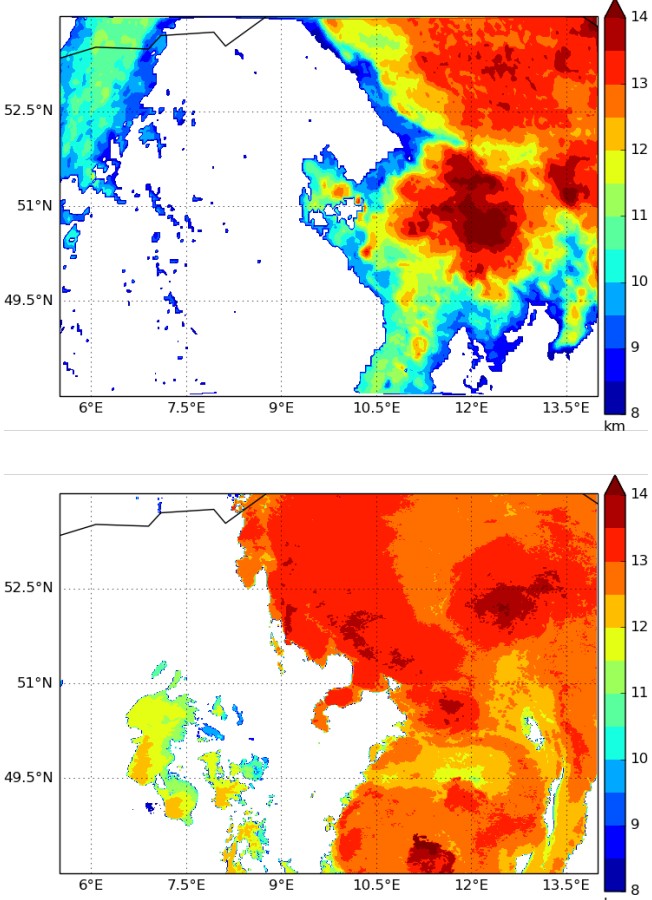

**Figure A1.** Cirrus temporal evolution as retrieved from the RAM-SES and satellite observations after 16 UTC on 4 July 2015. (Top) IWP, error estimates of the RAMSES retrieval are provided (cyan-shaded area). Orange-hatched bars indicate time periods for which CiPS IWP is flagged as compromised by an optical depth too high. (Center) Cloud vertical extent (grey-shaded area) and extinction-weighted cloud height as measured with RAMSES, and CiPS CTH. (Bottom) Optical depth. RAMSES data are corrected for multiple scattering. CiPS threshold optical depth is indicated (horizontal line).

**Figure A2.** Observed (top) and simulated (bottom) horizontal distribution of CTHs of ice clouds at 22 UTC for 4 July 2015 over Germany. CTHs below 8 km are not shown.

4 July 2015, when it might be exacerbated by an increased convective activity continuing into the night in the ICON-LEM simulation. Furthermore, the band of low ice clouds in the northwest of the domain (Fig. A2) is not captured by the model, which adds to the relative lack of simulated low CTHs.

## A3 Differences in initial and boundary data sets

The sensitivity simulations in Sect. 6.1 exhibit a significant spread in the cloud evolution and corresponding ice water path estimates and cloud top heights that is connected with the initial and boundary conditions provided by the driving models. It is therefore necessary to give a brief overview over the systematic differences in the initial and lateral boundary conditions provided by COSMO-DE, ICON-NWP and IFS analysis. For this comparison it needs to be pointed out that the analysis frequency of the different models varies (Tab. 2), favoring the COSMO-DE boundary conditions. Fig. A3 displays temperature, humidity and condensate profiles for the domain mean initial conditions (inset in each panel of Fig. A3) as well as the difference of ICON-LEM lbc1 (using ICON-NWP as forcing dataset) and ICON-LEM lbc2 (using IFS analysis) from the control simulation for five different times at the beginning of the simulation. The difference for 00 UTC reflects a domain mean difference over the full domain, whereas the subsequent +3, +6, +9 and +12 hour differences are mean differences over the 20 km nudging zone at the domain edges.

Focusing on the temperature profiles for both days (Fig. A3a and b) only minor differences for the IFS forced simulation (ICON-LEM lbc2) are apparent, with upper tropospheric temperatures in IFS by up to 1 K lower than compared to COSMO-DE. For the ICON-LEM lbc1 the higher

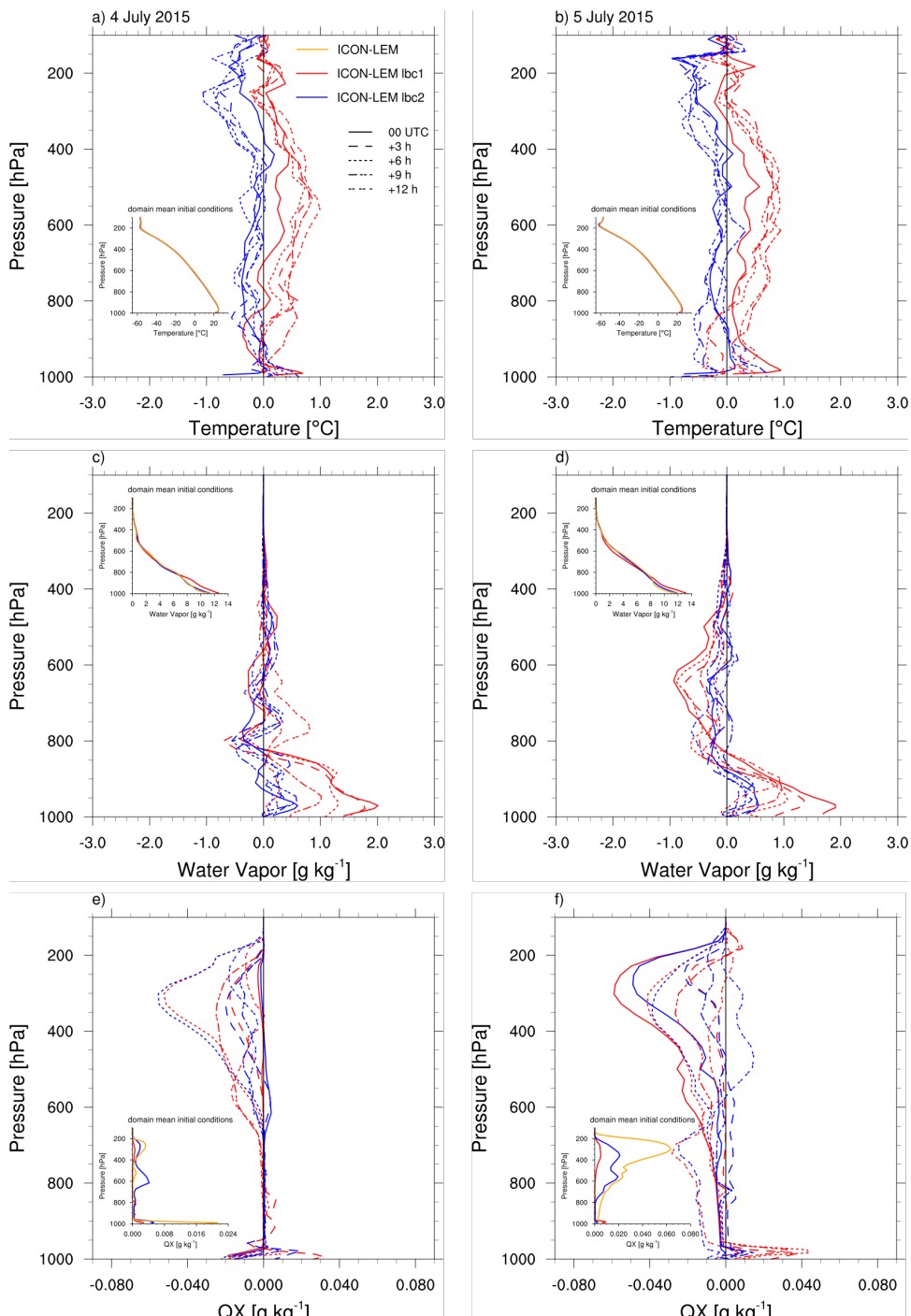

**Figure A3.** Initial and lateral boundary conditions used in Sect. 6.1 for temperature (top), water vapor (middle) and total condensate, comprising cloud water, rain, ice and snow, (abbreviated with QX; bottom). The vertical profiles show differences using ICON-NWP (ICON-LEM lbc1 - red) and IFS (ICON-LEM lbc2 - blue) analysis compared to the default simulation using analysis data provided by COSMO-DE. All forcing data sets are remapped onto the high resolution ICON-LEM DOM01 grid. Solid lines (00 UTC) display a mean difference over the full domain, whereas the subsequent +3, +6, +9 and +12 hour differences are averaged differences over the 20 km nudging zone at the domain edges, depicted with different line styles. The inlays show the mean initial (00 UTC) absolute profiles for the full domain including the control simulation (ICON-LEM - orange line). Vertical axis of the inlays are the same as in the difference plots. X-axis labels span for temperature the range between -60 °C to 10 °C, for water vapor 0 g kg$^{-1}$ to 14 g kg$^{-1}$ and for condensate 0 g kg$^{-1}$ to 0.024 g kg$^{-1}$.

initial temperatures (up to 1 K) close to the lowermost model layer below 950 hPa (solid red line) are most notable together with the slightly increased mid-tropospheric temperatures (between 300 and 700 hPa) in the lateral boundary data. Considering the moisture profiles the most striking difference can be found for ICON-LEM lbc1 and ICON-LEM with ICON-LEM lbc1 simulating significantly larger humidity below 800 hPa of up to 2 g kg$^{-1}$. On the one hand, a warmer boundary layer favors convection (if triggered) to be more vigorous and on the other hand, the higher humidity within the warmer troposphere causes higher condensation rates leading to increased latent heating. Although temperature and humidity discrepancies in the models are similar on both days, the impact on the simulations varies. On the 4th July, a thermally forced convective day, the impact of the varying initial and boundary data is larger than on the 5th July, where the impact is limited due to the large scale dynamical forcing connected with the frontal system. Additionally, the limited impact of the lower tropospheric moist bias on the 5th July may also be connected with the slight dry bias in the middle troposphere that leads to a decrease in humidity due to entrainment.

In addition to temperature and water vapor profiles the total condensate (QX), which is the sum of cloud water, rain, ice and snow, provided by the analysis data sets, is compared in Fig. A3e and f. The difference in initial conditions is minor for 4 July 2015, a day with little cloud condensate at the start of the day. Later during the day the boundary conditions in the ICON-LEM lbc1 and ICON-LEM lbc2 contain significantly lower condensate leading to a lower IWP, in closer agreement with observations. This difference is again reflected in the initial conditions of 5 July 2015 with the COSMO-DE forced simulation starting with significantly too much IWP and cloud cover in the upper troposphere (see Fig. 8b). Discrepancies in the lateral boundary conditions (+3 to +12 hours) could influence the upper tropospheric ice budget but should not be overinterpreted, because these fields refer only to the domain boundaries.

Given the significant differences in the forcing data based on COSMO-DE, ICON-NWP and IFS and resulting convective activity, a short overview of their data assimilation systems is paramount. While IFS (hybrid ensemble 4D-variational assimilation (4D-Var); Rabier et al., 2000) and COSMO-DE (local ensemble transform Kalman filter (LETKF); Hunt et al. (2007); Schraff et al. (2016)) both use well established and optimised data assimilation systems, ICON-NWP was first implemented on 20 January 2015 with a bare-bones 3D-Var system taken from the earlier global NWP model. During the year of 2015 multiple satellite and conventional observations were added and calibrated culminating in the 20 January 2016 implementation of a modern hybrid ensemble variational (EnVar) system. In this chapter's investigation of forcing impacts we specifically included the ICON-NWP forcing for a period in July 2015 shortly after first implementation because it provides a novel opportunity

to examine the possible range of uncertainty in forcing data sets.

## A4 Additional microphysical sensitivity simulations

The results of all microphysical experiments (Tab. A1) are summarized in condensed form in Tab. A2. Here we highlight only the values of the domain- and time-averaged liquid, respectively ice water path for cloud water (tqc), cloud ice (tqi), snow (tqs), graupel (tqg) and hail (tqh). Such simple statistics do nevertheless provide some insights. For example, the narrow ice particle size distribution leads to a slower ice sedimentation and, hence, a higher cloud ice water path (29 % increase compared to the control). The increased number of CCN leads to smaller cloud droplets, a suppression of warm rain formation and an increased lofting of water mass above the freezing level. Hence, cloud water is increased, rain water decreased, and cloud ice shows a strong increase of 46 % resp. 99 %. Interestingly, the precipitating ice categories of graupel and hail also show a significant reduction for increased CCN in these simulations. For a more detailed investigation and discussion of the impact of CCN in large-domain large-eddy simulations over Germany we refer to Costa-Surós et al. (2020). Compared to the other experiments, the assumptions regarding ice nuclei (IN) of experiments 12 to 14 have only a moderate impact on the simulation results, but the present-day aerosols (PDA) scheme leads to a significant increase in snow, graupel and hail, most notably in experiment 15, which assumes a significant contribution from organic IN. In the main text we focus on those microphysical experiments that lead to a decrease in cloud ice amount, which are experiments 3 and 4 with a modification of the cloud ice geometry, and experiment 10 with the increased sticking efficiency.

*Data availability.* Access to observational and model data sets used within this publication are provided under zenodo archive (Rybka, 2020)

*Author contributions.* UB, MK and HR created the conceptual design of this study. MK and LB selected the cases for suitability. Sensitivity simulations were planned and performed by HR, CM and AS. Satellite data were made available by AH, LB and JS and ground based data by JR and UG and evaluated against each other. HR, IA, UB and CM performed the analysis of the model simulations. The paper was jointly written by all authors.

*Competing interests.* The authors declare that they have no conflict of interest.

*Acknowledgements.* This work is funded by the research program "High Definition of Clouds and Precipitation for Advanc-

**Table A1.** Overview of the microphysical sensitivity experiments. In the SB scheme ice particles are characterized by power laws that relate the maximum dimension $D$ and the terminal fall velocity $v$ with particle mass $m$. The control simulation uses $D = 0.835m^{0.39}$ and $v = 27.7m^{0.216}$ for cloud ice where $D$ is in m, $v$ in m/s and $m$ in kg. For snow the control assumes $D = 5.13m^{0.5}$ and $v = 8.3m^{0.125}$. The particle size distribution is a generalized gamma distribution of the form $f(m) = Am^\nu \exp(-Bm^\mu)$, and the control run uses the shape parameters $\nu_i = 0$ and $\mu_i = 1/3$ for cloud ice and $\nu_s = 0$ and $\mu_s = 0.5$ for snow. $T_c$ is the cloud effective temperature.

| No. | simulation | description |
|-----|-----------|-------------|
| 1 | control | Control simulation with 625 m horizontal grid spacing (DOM01). |
| 2 | iceXmin | Reduction of minimum mean mass of cloud ice of $10^{-12}$ kg to $10^{-14}$ kg corresponding to a diameter of $4\mu$m. |
| 3 | hexPlate | Change cloud ice geometry to a plate-like habit with $D = 0.22m^{1/3.31}$ and a fall speed of $v = 41.9m^{0.26}$. |
| 4 | dendrite | Change cloud ice geometry to a dendrite-like habit with $D = 5.17m^{1/2.29}$ and a fall speed of $v = 11.0m^{0.21}$. |
| 5 | lightSnow | Change snow geometry to a low density snow with $D = 7.26m^{0.5}$ and a fall speed of $v = 3.6m^{0.1}$. |
| 6 | heavySnow | Change snow geometry to a high density snow with $D = 3.80m^{0.5}$ and a fall speed of $v = 7.5m^{0.1}$. |
| 7 | narrowIce | Narrow particle size distribution of cloud ice with $\nu_i = 2$ and $\mu_i = 1$. |
| 8 | narrowSnow | Narrow particle size distribution of snow with $\nu_s = 2$ and $\mu_s = 1$. |
| 9 | stickLFOlow | The sticking efficiency of $E_i = \exp(0.09T_c)$ is used for all ice-ice interactions. |
| 10 | stickLFOhigh | The sticking efficiency of $E_i = \exp(0.025T_c)$ is used for all ice-ice interactions. |
| 11 | stickLFOhigh2 | As exp. 10, but with $E_i = 0.01$ for $T_c < -40°$C. |
| 12 | Hande95 | Modified ice nucleation using the upper 95th percentile of the Spring conditions of Hande et al. (2015). |
| 13 | Hande05 | As exp. 13, but using the lowest 5th percentile (see Table 1 of Hande et al. (2015)). |
| 14 | PDA | Ice nucleation parametrized following PDA as specified in Seifert et al. (2012). |
| 15 | PDAorg | As exp. 14, but with additional organic particles, i.e. significantly more IN at around -10 °C. |
| 16 | 2xCCN | Twofold increase in CCN. |
| 17 | 4xCCN | Fourfold increase in CCN. |

**Table A2.** List of all microphysical sensitivity studies including domain-averaged bulk quantities of column-integrated cloud variables (in $\mathrm{g\,m^{-2}}$): cloud water (tqw), cloud ice (tqi), rain droplets (tqr), snow (tqs), graupel (tqg) and hail (tqh) and their relative difference (in %) to the ICON-LEM (DOM01) control simulation. All simulations in the microphysical studies were run with microphysics-radiation coupling turned on. Control no-mrc denotes a simulation where this coupling was turned off.

| simulation | tqc | rel. diff. | tqi | rel. diff. | tqr | rel. diff. | tqs | rel. diff. | tqg | rel. diff | tqh | rel. diff. |
|-----------|-----|-----------|-----|-----------|-----|-----------|-----|-----------|-----|-----------|-----|-----------|
| control | 50.98 | 0.0 | 110.20 | 0.0 | 53.21 | 0.0 | 23.47 | 0.0 | 151.06 | 0.0 | 12.05 | 0.0 |
| control no-mrc | 51.88 | – | 109.71 | – | 53.55 | – | 23.86 | – | 153.16 | – | 12.09 | – |
| iceXmin | 50.25 | -1.4 | 116.51 | 5.7 | 50.30 | -5.5 | 22.68 | -3.3 | 142.82 | -5.5 | 11.02 | -8.6 |
| hexPlate | 50.76 | -0.4 | 89.88 | -18.4 | 52.09 | -2.1 | 30.01 | 27.9 | 173.15 | 14.6 | 10.15 | -15.8 |
| dendrite | 49.03 | -3.8 | 92.02 | -16.5 | 52.38 | -1.6 | 29.84 | 27.1 | 164.19 | 8.7 | 9.81 | -18.6 |
| lightSnow | 50.48 | -1.0 | 109.43 | -0.7 | 53.27 | 0.1 | 27.86 | 18.7 | 152.35 | 0.9 | 12.43 | 3.2 |
| heavySnow | 51.22 | 0.5 | 108.38 | -1.6 | 52.48 | -1.4 | 21.31 | -9.2 | 149.28 | -1.2 | 11.80 | -2.0 |
| narrowIce | 50.19 | -1.5 | 142.13 | 29.0 | 50.26 | -5.5 | 21.48 | -8.5 | 144.99 | -4.0 | 10.22 | -15.2 |
| narrowSnow | 51.02 | 0.1 | 108.14 | -1.9 | 52.81 | -0.8 | 26.86 | 14.5 | 148.80 | -1.5 | 11.46 | -4.9 |
| stickLFOlow | 51.78 | 1.6 | 100.88 | -8.5 | 53.19 | 0.0 | 29.30 | 24.8 | 150.24 | -0.5 | 11.89 | -1.3 |
| stickLFOhigh | 56.43 | 10.7 | 19.55 | -82.3 | 57.78 | 8.6 | 21.66 | -7.7 | 156.92 | 3.9 | 14.35 | 19.2 |
| stickLFOhigh2 | 54.33 | 6.6 | 77.53 | -29.6 | 53.83 | 1.2 | 21.74 | -7.3 | 141.28 | -6.5 | 12.86 | 6.7 |
| Hande95 | 49.76 | -2.4 | 105.12 | -4.6 | 53.08 | -0.2 | 26.98 | 15.0 | 149.33 | -1.1 | 11.60 | -3.7 |
| Hande05 | 51.80 | 1.6 | 109.25 | -0.9 | 52.58 | -1.2 | 23.62 | 0.7 | 149.56 | -1.0 | 11.56 | -4.0 |
| PDA | 46.93 | -7.9 | 104.12 | -5.5 | 52.58 | -1.2 | 32.13 | 36.9 | 180.97 | 19.8 | 12.71 | 5.5 |
| PDAorg | 39.14 | -23.2 | 104.85 | -4.9 | 50.70 | -4.7 | 33.61 | 43.2 | 188.56 | 24.8 | 13.69 | 13.6 |
| 2xCCN | 59.92 | 17.5 | 161.26 | 46.3 | 45.98 | -13.6 | 25.68 | 9.4 | 136.12 | -9.9 | 10.42 | -13.5 |
| 4xCCN | 72.56 | 42.3 | 219.47 | 99.2 | 40.43 | -24.0 | 29.30 | 24.8 | 115.74 | -23.4 | 9.96 | -17.3 |

ing Climate Prediction" (HD(CP)²) of the BMBF (German Federal Ministry of Education and Research) under grant 01LK1505A, 01LK1505B, 01LK1505D and 01LK1505F. LB, IA and JS were funded by the DLR Klisaw project. We gratefully acknowledge the work of Ksenia Gorges and Rieke Heinze who performed the ICON-LEM control simulations and the computing time provided by the German Climate Computing Centre (DKRZ) on the HPC system Mistral and the Jülich Supercomputing Centre (JSC) on the HPC system JURECA. We also thank Bjorn Stevens and Wiebke Schubotz for initiating and coordinating the HD(CP)² project.

We thank two anonymous reviewers for helpful comments on earlier drafts of the manuscript.

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
