# Peer review of "The behavior of high-CAPE summer convection in large-domain large-eddy simulations with ICON"

_Atmospheric Chemistry and Physics, 2020_

## Referee Comment (RC1) · Anonymous Referee #1 · 30 Aug 2020

This is a well-organized and mostly well written paper describing an evaluation of summer convective events in large-eddy simulations over Germany using the ICON model. With respect to the difficult problem of predicting the evolution of convection, little new scientific insight is found. The conclusion from the findings in this study that this depends more heavily on the uncertainty of the large-scale dynamical state based on data assimilation rather than on microphysical parameters/schemes has long been established. Furthermore, the description of the sensitivity experiments with regards to the forcing datasets and their forecast impacts is rather vague and do not shed light on what aspects of the thermodynamic state are sufficiently or deficiently resolved in the various forcing datasets . The most significant aspect of the paper seems to be for doc-

umenting the ICON-LEM performance for the experiments performed here. The model is claimed to be a cutting edge tool for improving next-generation NWP models. The simulations are evaluated with a variety of ground-based measurements and satellite observations of cloud properties. The evaluations are reasonably thorough and the authors have done a nice job of assembling and describing the observational datasets, which are state of the art. While I can't comment on the model itself, the methods and data used in the evaluation are robust and presented in an informative way. Despite the somewhat limited significance of the study with respect to improving convective weather forecasting, I recommend that the manuscript could be published with minor revisions. While I am not a modeler, it seems to me that the manuscript could be improved by better describing the forcing datasets, their relative differences, and by better assessing and describing the impacts of these differences on the forecasts.

Other comments:

Line 227: Minnis et al 2008 would be more appropriate than the 2011 reference

Minnis, P., L. Nguyen, R. Palikonda, P. W. Heck, D. A. Spangenberg, D. R. Doelling, J. K. Ayers, W. L. Smith, Jr., M. M. Khaiyer, Q. Z. Trepte, L. A. Avey, F.-L. Chang, C. R. Yost, T. Chee, S. Sun-Mack, "Near-real time cloud retrievals from operational and research meteorological satellites", Proc. SPIE 7107, Remote Sensing of Clouds and the Atmosphere XIII, 710703 (13 October 2008); https://doi.org/10.1117/12.800344

Lines 286 and 988: Minnis 2020 should replace Minnis 2011

Minnis, P., S. Sun-Mack, Y. Chen, F.-L. Chang, C. R. Yost, W. L. Smith, Jr., P. W. Heck, R. F. Arduini, S. Bedka, Y.Yi, G. Hong, Z. Jin, D. Painemal, R. Palikonda, B. Scarino, D. A. Spangenberg, R. Smith, Q. Z. Trepte, P. Yang, and Y. Xie, 2020: CERES MODIS cloud product retrievals for Edition 4, Part 1: Algorithm changes. IEEE Trans. Geosci. Remote Sens., doi: 10.1109/TGRS.2020.3008866.

Line 558: there is a question from a co-author that should be addressed (on average

or that is peak reduction??)

Line 571: approx. should be approximately

Line 573-584 (and A2): it is stated: "Nevertheless, lower cloud top heights of up to 10 or 11 km are likely underestimated in the simulation. " I think that you mean the occurrence of lower heights? It isn't obvious from the text or figure 5 why you've reached this conclusion though. How do you know this isn't a problem with the observations? In fact, you state that the observations underestimate on lines 569-570. With the exception of CALIPSO, most other observing systems underestimate glaciated CTH and therefore would have higher frequencies of occurrence for the lower heights than actually occur. Thus, without other information, it seems to me that model CTH frequencies may in fact be more accurate than the observations. Unless I missed this in the text, it would be helpful to further support the contention that the lower CTH's are 'likely underestimated' in the simulations.

Line 592: seems like you could refer directly to 5.1.2 rather generally to 5.1

Lines 593-595: again. how do you know anvil heights are overestimated in the simulations?

Line 646: would it read better to say: "not captured in the default" ?

Pg 50: Fig A2 caption. Reconcile top/bottom with left/right figures

Line 828: "resp." ??

---

## Referee Comment (RC2) · Anonymous Referee #2 · 1 Oct 2020

Review of "The behavior of high-CAPE summer convection in large-domain large-eddy simulations with ICON" by Harald Rybka, Ulrike Burkhardt, Martin Köhler, Ioanna Arka, Luca Bugliaro, Ulrich Görsdorf, Ákos Horváth, Catrin I. Meyer, Jens Reichardt, Axel Seifert, and Johan Strandgren

Summary of paper:

The authors present results from 3 simulations of high-CAPE convective weather events over Germany with the ICON model at sub-km grid lengths. The simulations are evaluated in terms of ice cloud cover, ice water contents, and ice water paths against a range of ground-based and satellite-based retrievals. The model compares

well against the retrievals, although ice contents are vastly overestimated when graupel is included. Also, anvil cloud coverage and lifetime are overestimated. Sensitivity studies show greater sensitivity to the driving model (particularly in terms of the timing of initiation of convection) than to the microphysics parameters.

Review summary:

This is a very well-written paper with high-quality figures. While the paper is rather long (the introduction is already 3 pages), it is necessarily so due to its attempt to disentangle the sensitivity of the simulation due to the driving model from the sensitivity due to microphysics assumptions. The paper covers this comprehensively so and it is not obvious that its length (or number of figures) could be reduced significantly without losing useful results or discussion. There are no obvious flaws or concerns about the methodology or interpretation of the results and the only "major" comment relates to a request for further context. Given the quality of the presentation of the manuscript and the nature of the comments below, the recommendation is to accept with minor revisions.

Major comment:

This is only major in the sense that it requires a small amount of work, but given the importance of CAPE for this study it would provide the reader with further confidence in the simulations if the authors presented a brief evaluation of the thermodynamic profiles for the three cases. Of course, such observed profiles may only be available at specific times and will require some cherry-picking of locations and times in relation to the convective activity. Nevertheless, the thermodynamic profiles would provide adequate context of potential model biases in (1) cloud top height and (2) timing of convection. Both of these are of interest in the sensitivity analysis comparing different driving models, so that it's worth revisiting the thermodynamic profiles in that section as well.

Minor comments:

[Figure]

Line 45-48: To what extent can cloud resolving simulations be considered "truth"? Please include a few references that have explored differences between models and sensitivity for a given model (e.g. to resolution).

Line 71-73: The list of previous studies and their relevant topics is nice to see, but it does diminish the impact of the present paper. Please add 1-2 sentences describing what processes can be explored with these case studies specifically. What is so unique about a high CAPE environment that the prior studies didn't explore?

Line 77: Remove parenthesis around the reference.

Line 90: The grid spacings for the driving models do not match the value cited in Table 2.

Line 140: "been reported for this day" – A reference would be great, more out of curiosity than out of scientific necessity.

Line 141-142: "convective inhibition" – Should this be explored in relation to the driving model and or temporal differences?

Line 190-192: Shorten this to (e.g.): "In addition to the three days of interest described in Section 2, we further. . ."

Line 200: Is the forcing from these other two driving models still 3-hourly? Please specify.

Line 203-206: The ice particle habits are only mentioned towards the end of Section 4. It is worth introducing them here and indicate which habits are directly affected by the change to ice particle geometry and fall speed (e.g. is "snow" affected or not?).

Line 234: "next sections" – "next sub-sections"

Line 237: "Lindenberg" – It would be helpful to the reader not familiar with Germany to indicate the location of this site and the other two in Figure 1.

[Figure]

Line 244: "new method" – If not against ACP policy, it would be helpful to cite manuscript in preparation here, or make a stronger statement indicating that this method was developed "in conjunction" or "in parallel" with this study.

Line 251: "these wavelengths" – Presumably this only refers to the 35 GHz radar, so should be singular. Also, it is worth mentioning that the retrieval suffers from attenuation in profiles with heavy precipitation (as clearly evident in Figure 2 and mentioned in the caption).

Line 312: "both day and night" – Are there any previous studies exploring how seamless this retrieval is? Are there differences in errors and detection efficiency between day and night? It would be helpful to add these.

Line 316: The wavelengths are the same as APICS. Has there been an intercomparison study of these retrievals? It is helpful to cite known differences.

Line 333: As above, any known differences between these retrievals would be useful to cite now.

Line 385-394: While this is all correct, this text is rather superfluous as the CloudSat/CALIPSO retrievals are not used directly in this study. The text can be removed without loss of understanding.

Line 404: "radar/lidar-based" – Preferred "radar/lidar-trained" as the actual measurements are not radar or lidar.

Line 409-410: "and space" – How is the error in space considered in this analysis? All one can do with the grid-point comparison is assume that there is only an error in time.

Line 464: "RAMSES observations" – "retrievals"

Line 465-471: "well reproduced" – I appreciate that not everything needs to be shown, but it's worth knowing which observational data rainfall was compared against.

Line 472-485: This evaluation against RAMSES seems to completely ignore earlier

statements that RAMSES is primarily reliable in the lower reaches of the cloud and that it can suffer from "strong signal attenuation" (Line 462). It would be worth considering the potential that RAMSES significantly underestimates IWC in thick clouds, as suggested by the authors themselves.

Line 558: "(on average or that is peak reduction??)" – This is a good question!

Line 641-643: The authors mention the "wet moisture bias", but are there generally differences in the thermodynamic profiles between the simulations run from the different driving models? Particularly in terms of convective inhibition?

Line 683-684: Please specify the location of the field campaign CRYSTAL-FACE.

Line 692-712: Is there a significant change in latent heat release from these sensitivity runs due to the increase in riming that could affect cloud dynamics and hence duration and anvil extent? A brief comment in the text would be appreciated.

Line 750: "deficiencies in the microphysical scheme" – Is there any chance that there could be deficiencies in the radiative effects of the anvil cloud? The effects of radiative processes and latent heating have been shown to affect cloud lifetime e.g. Gaparini et al. (2019)

Gasparini, B., Blossey, P.N., Hartmann, D.L., Lin, G. and Fan, J., 2019. What Drives the Life Cycle of Tropical Anvil Clouds?. Journal of Advances in Modeling Earth Systems, 11(8), pp.2586-2605.

Line 767-768: How do the authors envisage constraining the graupel estimates? Would weather radar observations help or revisiting campaigns such as COPS (or proposing a new campaign!)? In other words: What is needed to improve the representation of graupel?

Line 793-798: The concluding remarks focus on future satellite missions, but some words on ground-based would be appreciated here, too. The comparison against the CloudNet retrievals looks promising, even if only briefly considered in the paper. A

consideration of more cases would eventually allow a statistical evaluation against the CloudNet sites. Separately, there may be more complementary information from the Julich multi-instrument site that could be exploited.

Figure 2: Please specify in the caption the reason for the regular failure of Cloudnet retrievals for Julich – is there something specific about the measurements at those times? Regarding "Temporal retrievals", please specify the temporal frequency, e.g. every 5 minutes.

Figure 4: Please specify in the caption the meaning of tqi and tqf.

Figure 6: The x-axis of the second panel says "CLCH" instead of "ICC".

Figure 9: It would be helpful to also specify in the caption which categories are directly affected by the change in microphysics parameters.

[Figure]

---

## Author Comment (AC1) · 14 Dec 2020

**Authors response to review**

Review comments pasted in black.
Author response in blue. New line references refer to the revised manuscript including newly added paragraphs/sentences set in italic font.

**Review No. 1**
This is a well-organized and mostly well written paper describing an evaluation of summer convective events in large-eddy simulations over Germany using the ICON model. With respect to the difficult problem of predicting the evolution of convection, little new scientific insight is found. The conclusion from the findings in this study that this depends more heavily on the uncertainty of the large-scale dynamical state based on data assimilation rather than on microphysical parameters/schemes has long been established. Furthermore, the description of the sensitivity experiments with regards to the forcing datasets and their forecast impacts is rather vague and do not shed light on what aspects of the thermodynamic state are sufficiently or deficiently resolved in the various forcing datasets.

The most significant aspect of the paper seems to be for documenting the ICON-LEM performance for the experiments performed here. The model is claimed to be a cutting edge tool for improving next-generation NWP models. The simulations are evaluated with a variety of ground-based measurements and satellite observations of cloud properties. The evaluations are reasonably thorough and the authors have done a nice job of assembling and describing the observational datasets, which are state of the art. While I can't comment on the model itself, the methods and data used in the evaluation are robust and presented in an informative way. Despite the somewhat limited significance of the study with respect to improving convective weather forecasting, I recommend that the manuscript could be published with minor revisions. While I am not a modeler, it seems to me that the manuscript could be improved by better describing the forcing datasets, their relative differences, and by better assessing and describing the impacts of these differences on the forecasts.

We would like to thank the reviewer for the helpful report. The manuscript has been revised according to the comments.

Please note, that in response to comments of reviewer 2 we have added a sub-section (NEW: section 5.1 - Evaluation of simulated temperature profiles with radiosonde data), and in response to both reviewers an appendix (NEW: Appendix 3 - Differences in initial and boundary data sets). These newly added paragraphs help to gain more insights into the triggering and properties of convection and its representation within the model and to which degree differences in the simulated convection depend on the forcing data sets.

Other comments:
Line 227: Minnis et al 2008 would be more appropriate than the 2011 reference.
Reference has been changed.

Lines 286 and 988: Minnis 2020 should replace Minnis 2011.

Reference has been changed.

Line 558: there is a question from a co-author that should be addressed (on average or that is peak reduction??).
The text has been edited to specify that a domain average tqf reduction at times of peak ice water path is affected by applying an upper threshold to the simulated frozen water path (Line 611-613). As explained within the paragraph only up to 3.5 % of the model grid points are affected by applying the threshold at time of peak ice water path. Therefore the tqf domain average is affected when having its maximum values during the day.

Line 571: approx. should be approximately.
Done

Line 573-584 (and A2): it is stated: "Nevertheless, lower cloud top heights of up to 10 or 11 km are likely underestimated in the simulation. " I think that you mean the occurrence of lower heights? It isn't obvious from the text or figure 5 why you've reached this conclusion though. How do you know this isn't a problem with the observations? In fact, you state that the observations underestimate on lines 569-570. With the exception of CALIPSO, most other observing systems underestimate glaciated CTH and therefore would have higher frequencies of occurrence for the lower heights than actually occur. Thus, without other information, it seems to me that model CTH frequencies may in fact be more accurate than the observations. Unless I missed this in the text, it would be helpful to further support the contention that the lower CTH's are 'likely underestimated' in the simulations.
Yes, we mean the occurrence of lower heights. CTH is derived from the CiPS algorithm. Strandgren et al. 2017 compare CTH estimated by CiPS with CALIOP CTH and show in their Fig. 10 that at latitudes relevant for the Germany domain CiPS retrieves almost bias free CTHs for ice cloud tops located between approx. 8 and 11 km, while it tends to underestimate CTHs that are higher than 11 km and to overestimate CTHs that are lower than 8 km. Thus, we are confident that the probability of occurrence of CTHs in the range 8–11 km is well retrieved by CiPS and that the model tends to underestimate these CTHs. We have added an explanation to the manuscript at this point and an additional sentence in Sect. 4.3.

Section 4.3 - Line 282-286: *The CTH retrieved by CiPS has an average error of 10 % or less for cirrus clouds with a top height greater than 8 km, again with respect to CALIOP observations over the entire MSG disk. When looking at the geographic distribution of CTH accuracy of CiPS versus CALIOP, it turns out that the CiPS neural network has a mean percentage error very close to zero in Germany for ice clouds located between 8 and 11 km. For lower clouds, CiPS tends to overestimate and for higher clouds to underestimate CTH.*
Section 5.2.2 - Line 625-627: *Validation against CALIOP (Strandgren et al., 2017a, Fig. 10) shows that at German latitudes CiPS retrieves almost bias free CTHs for ice*

*cloud tops located between approx. 8 and 11 km, while it tends to underestimate CTHs that are higher than 11 km and to overestimate CTHs that are lower than 8 km.*

Strandgren, J., Bugliaro, L., Sehnke, F., and Schröder, L.: Cirrus cloud retrieval with MSG/SEVIRI using artificial neural networks, Atmospheric Measurement Techniques, https://doi.org/10.5194/amt-10-3547-2017

Line 592: seems like you could refer directly to 5.1.2 rather generally to 5.1
Done

Lines 593-595: again. how do you know anvil heights are overestimated in the simulations?
In general, the characteristics of the CIPS accuracy is given above. In these particular lines however the focus is on anvil edges as they are discussed in Appendix A2. Here the CiPS observations show that the thunderstorm cloud has highest CTHs in the "centre", i.e. in the convective part, and that CTH decreases towards the cloud edge (the blue colours in Fig. A2, top). The model, Fig. A2, bottom, also shows peak CTH in the inner part of the cloud, but CTH is overestimated towards the cloud edges. We have added a sentence in the manuscript.

Section 5.3 - Line 652-655: *As shown in Appendix 2, the CiPS observations show that the thunderstorm cloud has highest CTHs in and around the convective core and that CTH decreases towards the cloud edge (the blue colours in Fig. A2, top). The model, bottom panel in Fig. A2, also shows CTH peaks in the inner part of the cloud, but it lacks the realistic simulation of the CTH distribution towards the cloud edges.*

Line 646: would it read better to say: "not captured in the default"?
Has been changed accordingly.

Pg 50: Fig A2 caption. Reconcile top/bottom with left/right figures
The images will be displayed on top of each other in following documents and within the final document.

Line 828: "resp." ??
resp. = respectively - This abbreviation has been changed.

**Review No. 2**

Review summary: This is a very well-written paper with high-quality figures. While the paper is rather long (the introduction is already 3 pages), it is necessarily so due to its attempt to disentangle the sensitivity of the simulation due to the driving model from the sensitivity due to microphysics assumptions. The paper covers this comprehensively so and it is not obvious that its length (or number of figures) could be reduced significantly without losing useful results or discussion. There are no obvious flaws or concerns about the methodology or interpretation of the results and the only "major" comment relates to a request for further context. Given the quality of the presentation of the manuscript and the nature of the comments below, the recommendation is to accept with minor revisions.

We would like to thank the reviewer for the detailed comments and good suggestions. The manuscript has been revised according to the referee's comments.

Major comment: This is only major in the sense that it requires a small amount of work, but given the importance of CAPE for this study it would provide the reader with further confidence in the simulations if the authors presented a brief evaluation of the thermodynamic profiles for the three cases. Of course, such observed profiles may only be available at specific times and will require some cherry-picking of locations and times in relation to the convective activity. Nevertheless, the thermodynamic profiles would provide adequate context of potential model biases in (1) cloud top height and (2) timing of convection. Both of these are of interest in the sensitivity analysis comparing different driving models, so that it's worth revisiting the thermodynamic profiles in that section as well.

Thank you for this suggestion. We have added a new section (NEW 5.1 - Evaluation of simulated temperature profiles with radiosonde data) to compare the simulation with radiosonde data. We have focused this analysis on the stability of the atmosphere studying temperature and dew-point temperature profiles at two locations (marked in Figure 1). Furthermore, CAPE and CIN have been examined and results have been incorporated in section 2 as well. We selected radiosonde profiles fitting to the times of day shown in Figure 1 or right on the edge of convective activity. The comparison shows that ICON-LEM provides accurate results concerning thermodynamic states for the selected high-CAPE convective cases.

Additionally we have added an appendix (NEW: Appendix 3 - Differences in inital and boundary data sets), analyzing the different forcing data sets in terms of temperature, water vapor and total condensate profiles. In section 6.1 we use this information to argue why the simulations using different initial and boundary data result in different representations of the convective systems regarding the evolution of IWP and CTH.

Minor comments: Line 45-48: To what extent can cloud resolving simulations be considered "truth"? Please include a few references that have explored differences between models and sensitivity for a given model (e.g. to resolution).

We have expanded the introduction covering more literature in the areas of parameterization development and improvements of simulations due to increased resolution but we

kept it short since projects focusing on those differences (CASCADE and COPE) are mentioned below in the introduction - Below is the text added in the Introduction:

Introduction - Line 45-52: *Cloud resolving, as opposed to convection permitting, modeling is seen at present as a way of developing and testing parameterizations for low resolution models (Guichard and Couvreux, 2017; Gentine et al., 2018; Derbyshire et al. 2004), which require a detailed evaluation of the simulated cloud cover, water content, and cloud top heights. Cloud resolving modeling has been shown to lead to significant improvements in the representation of cloud and precipitation processes (e.g. Stevens et al., 2020; Khairoutdinov et al. 2009) and the continuing development of the models will improve the inclusion of small-scale couplings such as between turbulence and microphysics and with the land-surface (Guichard and Couvreux, 2017). Moreover, these models are starting to be run globally and have the potential to overcome the persistent problems of low-resolution models (Tomita et al., 2005; Satoh et al., 2019; Stevens et al. 2019).*

Line 71-73: The list of previous studies and their relevant topics is nice to see, but it does diminish the impact of the present paper. Please add 1-2 sentences describing what processes can be explored with these case studies specifically. What is so unique about a high CAPE environment that the prior studies didn't explore?
Thank you for the comment. We have now changed the text pointing out the differences in the studies concerned with convection (Line 73-79). In particular we detail that several papers look at measures of convective organization and aggregation, one study looks at the impact of soil moisture on convection and the Senf et al. (2018) paper studies spatial statistics of tropical clouds in a lower resolved model version.

Senf, F., D. Klocke, and M. Brueck, 2018: Size-Resolved Evaluation of Simulated Deep Tropical Convection. Mon. Wea. Rev., 146, 2161–2182, https://doi.org/10.1175/MWR-D-17-0378.1.

Line 77: Remove parenthesis around the reference.
Done

Line 90: The grid spacings for the driving models do not match the value cited in Table 2.
The grid spacings have been corrected!

Line 140: "been reported for this day" – A reference would be great, more out of curiosity than out of scientific necessity.
A reference has been added referring to the ESWD (European Severe Weather Database; https://eswd.eu/cgi-bin/eswd.cgi) homepage including its literature reference (Dotzek et al., 2009a).

Dotzek, N., P. Groenemeijer, B. Feuerstein, and A. M. Holzer, 2009a: Overview of

ESSL's severe convective storms research using the European Severe Weather Database ESWD. Atmos. Res., 93, 575-586

Line 141-142: "convective inhibition" – Should this be explored in relation to the driving model and or temporal differences?
In conjunction with the newly added section 5.1 (see response to major comment), we have provided information on CAPE and CIN based on single locations, which are strongly affected by convection.

Line 190-192: Shorten this to (e.g.): "In addition to the three days of interest described in Section 2, we further..."
We have shortened this paragraph and removed the repetition of all convective days and abbreviations.

Line 200: Is the forcing from these other two driving models still 3-hourly? Please specify.
The temporal update of the lateral boundary forcing is the same for all three cases. The only difference for IFS and ICON forcing is that forecast fields for 3, 6 and 9 hours after 00 UTC and 12 UTC are used as "quasi-analysis" fields. This has been explicitly added in the text (Line 206-208). Table 3 has been expanded with an additional column (frequency of analysis). In between analysis time steps hourly forecasts are available as boundary conditions.

Line 203-206: The ice particle habits are only mentioned towards the end of Section 4. It is worth introducing them here and indicate which habits are directly affected by the change to ice particle geometry and fall speed (e.g. is "snow" affected or not?).
We agree, that the introduction of habits in terms of type or geometry of a frozen hydrometeor (cloud ice, snow, graupel or hail) should be mentioned in Section 2. The revised version includes this correction (Line 212-214). The direct impact of changes within the microphysical scheme for the three selected sensitivity experiments has already been stated within this paragraph, i.e.: the ice crystal geometries of hexagonal plates and dendrites have a direct impact on cloud ice, whereas the high sticking efficiency simulation affect all habits except hail due to the ice collision (i.e.: snow-snow, cloud ice-cloud ice, snow-cloud ice, graupel-snow).

Line 234: "next sections" – "next sub-sections"
Done

Line 237: "Lindenberg" – It would be helpful to the reader not familiar with Germany to indicate the location of this site and the other two in Figure 1.
The location of all ground-based observational sites have been marked in Figure 1.

Line 244: "new method" – If not against ACP policy, it would be helpful to cite manuscript in preparation here, or make a stronger statement indicating that this

method was developed "in conjunction" or "in parallel" with this study.

The new method will be published in the future. We have edited the sentence to indicate that this was done in conjunction with our study (Line 255).

Line 251: "these wavelengths" – Presumably this only refers to the 35 GHz radar, so should be singular. Also, it is worth mentioning that the retrieval suffers from attenuation in profiles with heavy precipitation (as clearly evident in Figure 2 and mentioned in the caption).

We have corrected this mistake and added a short comment on the lower detection capability during heavy precipitation.

Section 4.2 - Line 263-264: *Only in situations with strong precipitation the attenuation is higher and thus the cloud detection capability lower.*

Line 312: "both day and night" – Are there any previous studies exploring how seamless this retrieval is? Are there differences in errors and detection efficiency between day and night? It would be helpful to add these.

Nighttime retrievals are generally more uncertain due to the loss of information contained in the solar reflectance channel. The nighttime algorithm also has a tendency to favor ice-phase retrievals (nighttime phase is biased towards ice clouds). We have added two further studies (Minnis et al., 2020 and Yost et al., 2020) and rephrased that paragraph.

Section 4.5 - Line 324-331: *SatCORPS is the only geostationary retrieval used here that provides IWP during both day and night for thin and thick ice clouds. […] Nighttime retrievals are inherently more uncertain due to the reduced information content resulting from the lack of the solar reflectance channel (Minnis et al., 2020) and the nighttime algorithm has a tendency to favor ice-phase retrievals (Yost et al., 2020). The pixel-level 15-minute temporal resolution SEVIRI SatCORPS data were obtained from NASA Langley Research Center (http://satcorps.larc.nasa.gov, last access: 15 April 2019).*

P. Minnis et al., "CERES MODIS Cloud Product Retrievals for Edition 4–Part I: Algorithm Changes," in IEEE Transactions on Geoscience and Remote Sensing, doi: 10.1109/TGRS.2020.3008866.

C. R. Yost, P. Minnis, S. Sun-Mack, Y. Chen and W. L. Smith, "CERES MODIS Cloud Product Retrievals for Edition 4–Part II: Comparisons to CloudSat and CALIPSO," in IEEE Transactions on Geoscience and Remote Sensing, doi: 10.1109/TGRS.2020.3015155.

Line 316: The wavelengths are the same as APICS. Has there been an intercomparison study of these retrievals? It is helpful to cite known differences.

We have added two sentences at the end of section 4.6, pointing out that SEVIRI CPP and SEVIRI APICS are very similar, but CPP retrieves smaller IWPs due to its different assumed ice habit and lower $\tau$ truncation threshold.

Older versions of CPP and APICS have been compared indirectly in Bugliaro et al. 2011. There as well it shows that CPP produces lower values of optical thickness and IWP than APICS.

Section 4.6 - Line 341-344: *Due to the different assumed ice habit and smaller $\tau$ truncation threshold, SEVIRI CPP retrieves smaller IWP values than SEVIRI APICS, although the algorithms are otherwise very similar. Older versions of CPP and APICS also show in Bugliaro et al. (2011) that they provide similar results, with again CPP producing lower values of optical thickness and IWP than APICS.*

Bugliaro, L., Zinner, T., Keil, C., Mayer, B., Hollmann, R., Reuter, M., and Thomas, W.: Validation of cloud property retrievals with simulated satellite radiances: a case study for SEVIRI, Atmospheric Chemistry and Physics, https://doi.org/10.5194/acp-11-5603-2011

Line 333: As above, any known differences between these retrievals would be useful to cite now.
A comparison has been performed by Benas et al. (2017). They found lower SEVIRI CPP IWPs than MODIS IWPs due to lower ice effective radius for CPP. We have added a sentence about this at the end of section 4.7.
An upcoming work is the comparison of SEVIRI CPP/APICS, MODIS, SPARE-ICE and RAMSES IWP.

Section 4.7 - Line 357-359: *Benas et al. (2017) compared SEVIRI CPP and MODIS retrievals. They found lower CPP IWPs than MODIS IWPs, similar to our observations (see Fig. 5), mainly caused by lower CPP ice effective radius values.*

Benas, N., Finkensieper, S., Stengel, M., van Zadelhoff, G.-J., Hanschmann, T., Hollmann, R., and Meirink, J. F.: The MSG-SEVIRI-based cloud property data record CLAAS-2, Earth Syst. Sci. Data, 9, 415–434, https://doi.org/10.5194/essd-9-415-2017, 2017

Line 385-394: While this is all correct, this text is rather superfluous as the CloudSat/CALIPSO retrievals are not used directly in this study. The text can be removed without loss of understanding.
We agree with the reviewer that most of the text can be removed. We have reorganized the paragraph and deleted most of the text. The only point kept, is the sensitivity of CloudSat/CALIPSO to the large hydrometeors and consequently SPARE-ICE, that was trained on them.

Line 404: "radar/lidar-based" – Preferred "radar/lidar-trained" as the actual measurements are not radar or lidar.
Done

Line 409-410: "and space" – How is the error in space considered in this analysis? All one can do with the grid-point comparison is assume that there is only an error in time. Considering the spatial field can indicate if a location error could be responsible. If the field is inhomogeneous and in the comparison a feature is simulated similar to the observed one, then a location error could be responsible for the error. We added the following text:

Section 4.10 - Line 422-424: *Furthermore, we take into account the neighboring grid points since differences between observations and simulations may be easily explained in case of inhomogeneities. This comparison approach is intended to provide an assessment of the model simulation error considering potential temporal or spatial displacements.*

Line 464: "RAMSES observations" – "retrievals"
In this sentence "RAMSES observations" is referring to measurements of cloud optical geometries which are direct measurements. We would like to keep this term.

Line 465-471: "well reproduced" – I appreciate that not everything needs to be shown, but it's worth knowing which observational data rainfall was compared against.
The intensity of different precipitation events have been evaluated by attenuated backscatter profiles of the ceilometer observations in Lindenberg and rain gauge measurements confirmed these results. This has been included in the text.

Section 5.2.1 - Line 519-521: *Simulated precipitation intensity was compared to estimates from attenuated backscatter profiles from ceilometer observations which were confirmed by rain gauge measurements.*

Line 472-485: This evaluation against RAMSES seems to completely ignore earlier statements that RAMSES is primarily reliable in the lower reaches of the cloud and that it can suffer from "strong signal attenuation" (Line 462). It would be worth considering the potential that RAMSES significantly underestimates IWC in thick clouds, as suggested by the authors themselves.
The effect of a strong signal attenuation is first of all referring to the decline in CTH. Direct measurement of IWC is only available for lower parts of the cloud due to the limitations stated above. To overcome this limitation a new method has been applied and described in Appendix A1. Using the cross-polarized backscatter coefficient as a proxy, IWC and IWP can be derived under all conditions. We did not suggest an underestimation of IWC by RAMSES.

Line 558: "(on average or that is peak reduction??)" – This is a good question!
We have changed the text specifying that a tqf average reduction at peak times is affected by applying an upper threshold to the simulated frozen water path.

Line 641-643: The authors mention the "wet moisture bias", but are there generally

differences in the thermodynamic profiles between the simulations run from the different driving models? Particularly in terms of convective inhibition?

We have added an appendix (NEW: Appendix 3 - Differences in initial and boundary data sets) analyzing differences in the forcing data sets. We refer to our response concerning the major comments in the beginning.

Line 683-684: Please specify the location of the field campaign CRYSTAL-FACE.

"in Florida 2002" has been added to the text.

Line 692-712: Is there a significant change in latent heat release from these sensitivity runs due to the increase in riming that could affect cloud dynamics and hence duration and anvil extent? A brief comment in the text would be appreciated.

We have not seen a significant change in the latent heat release or the cloud dynamics. This would require a more detailed investigation, e.g., using cloud tracking to identify the life cycle of individual convective cells. This is beyond the scope of the paper, unfortunately, but would be a very interesting study. We have added this statement within this paragraph.

Section 6.2 - Line 767-769: *Additionally, latent heat release or cloud dynamics did not change significantly. In order to investigate this in more detail, a cloud tracking algorithm could unveil new insights in the life cycle of individual convective cells, which is beyond the scope of this paper.*

Line 750: "deficiencies in the microphysical scheme" – Is there any chance that there could be deficiencies in the radiative effects of the anvil cloud? The effects of radiative processes and latent heating have been shown to affect cloud lifetime e.g. Gaparini et al. (2019)

We agree that a large uncertainty is connected with the coupling of ice clouds and radiation stemming in particular from uncertainties in ice crystal habit and the different hydrometeor types and particle sizes. The coupling between radiation and cloud properties was improved regarding particle sizes for the microphysical perturbation experiments. Comparing the standard high resolution simulation and the control simulation for the microphysical experiments we could not find significant changes. Nevertheless, we agree that the radiative effects of anvils are difficult to describe because of vertical resolution as well as the optical properties of ice, snow and graupel. We are currently working on improving the consistency between the particle size spectrum used in the microphysical scheme and their radiative properties including an extension to large crystal sizes.

We take into account the possibility of deficiencies in cloud-radiation interactions of the anvil cloud influencing the too homogeneous cloud top heights within the simulation (Line 816-817).

Line 767-768: How do the authors envisage constraining the graupel estimates? Would weather radar observations help or revisiting campaigns such as COPS (or proposing a new campaign!)? In other words: What is needed to improve the representation of

graupel?

Measuring the degree of riming of snow and graupel would indeed be key to this. There are some recent developments using video disdrometer or vertically pointing radar, see e.g. Praz et al., 2017, Kneifel and Moisseev, 2020 or Ori et al., 2020.

So, yes, a measurement campaign aiming at such quantities could be very helpful to shed some light on these issues. Even operational weather radars could maybe be used if they include a vertically pointing mode (birdbath scan) in their scan strategy. We have included these statements in our discussion.

Conclusions - Line 835-837: *Measuring the degree of riming would be key to constrain graupel estimates. Recent developments using a video disdrometer (Praz et al., 2017) or vertically pointing radar (Kneifel and Moisseev, 2020; Ori et al.) shed some light on this issue.*

Praz, C., Roulet, Y.-A., and Berne, A.: Solid hydrometeor classification and riming degree estimation from pictures collected with a Multi-Angle Snowflake Camera, Atmos. Meas. Tech., 10, 1335–1357, https://doi.org/10.5194/amt-10-1335-2017, 2017.

Kneifel, S., and D. Moisseev, 2020: Long-Term Statistics of Riming in Nonconvective Clouds Derived from Ground-Based Doppler Cloud Radar Observations. J. Atmos. Sci., 77, 3495–3508, https://doi.org/10.1175/JAS-D-20-0007.1.

Ori, D, Schemann, V, Karrer, M, et al. Evaluation of ice particle growth in ICON using statistics of multi-frequency Doppler cloud radar observations. QJR Meteorol Soc. 2020; 1– 20. https://doi.org/10.1002/qj.3875

Line 793-798: The concluding remarks focus on future satellite missions, but some words on ground-based would be appreciated here, too. The comparison against the CloudNet retrievals looks promising, even if only briefly considered in the paper. A consideration of more cases would eventually allow a statistical evaluation against the CloudNet sites. Separately, there may be more complementary information from the Julich multi-instrument site that could be exploited.

Indeed, a more statistical evaluation including multiple CloudNet sites would be a nice follow-up study of the Illingworth et al., 2007 intercomparison. So far, only single observational sites have been used to evaluate this model version (Schemann and Ebell, 2019; Nomokonova et al., 2020). Further specific ICON-LEM model evaluation has been performed over JOYCE (Jülich Observatory for Cloud Evolution), see e.g. Marke et al. 2018, Ori et al. 2020.

Conclusions - Line 862-865: *A statistical intercomparison using multi-site Cloudnet information (following the study of Illingworth et al. (2007)) would allow a more comprehensive evaluation for future hectoscale NWP models, which has only been performed over single locations so far (Nomokonova et al., 2019; Schemann and Ebell, 2020).*

Schemann, V. and K. Ebell, 2020: Simulations of mixed-phase clouds with the ICON-LEM in the complex Arctic environment around Ny–Ålesund, Atmos. Chem. Phys., 20, 475–485, https://doi.org/10.5194/acp-20-475-2020

Nomokonova, T., K. Ebell, U. Löhnert, M. Maturilli, C. Ritter, and E. O'Connor, 2019: Statistics on clouds and their relation to thermodynamic conditions at Ny-Ålesund using ground-based sensor synergy, Atmos. Chem. Phys., 19, 4105-4126, doi:10.5194/acp-19-4105-2019

Marke, T., S. Crewell, V. Schemann, J. H. Schween, and M. Tuononen, 2018: Long-Term Observations and High-Resolution Modeling of Midlatitude Nocturnal Boundary Layer Processes Connected to Low-Level Jets. J. Appl. Meteor. Climatol., 57, 1155–1170, https://doi.org/10.1175/JAMC-D-17-0341.1.

Figure 2: Please specify in the caption the reason for the regular failure of Cloudnet retrievals for Julich – is there something specific about the measurements at those times? Regarding "Temporal retrievals", please specify the temporal frequency, e.g. every 5 minutes.
We have added the temporal frequency (30 s) in the caption of Figure 2 and included the reason for the periodically reoccurring retrieval gaps. The latter is a consequence of a radar scan every hour in which the antenna is not vertically pointed and thereby a Cloudnet retrieval is not possible.

Figure 4: Please specify in the caption the meaning of tqi and tqf.
A short description of tqi and tqf has been added to the caption and a reference to the sub-section for further explanation.

Figure 6: The x-axis of the second panel says "CLCH" instead of "ICC".
The label of the x-axis has been corrected.

Figure 9: It would be helpful to also specify in the caption which categories are directly affected by the change in microphysics parameters.
Only the cloud ice category is directly affected by these changes. This is already mentioned in the caption, but we have changed "ice geometries" to "cloud ice geometries" to make this more clear.